# Accessing the main-group metal formyl scaffold through CO-activation in beryllium hydride complexes

Terrance J. Hadlington [1✉] & Tibor Szilvási[2]

Carbon monoxide (CO) is an indispensable C1 building block. For decades this abundant gas has been employed in hydroformylation and Pausen-Khand catalysis, amongst many related chemistries, where a single, non-coupled CO fragment is delivered to an organic molecule. Despite this, organometallic species which react with CO to yield C1 products remain rare, and are elusive for main group metal complexes. Here, we describe a range of amido-beryllium hydride complexes, and demonstrate their reactivity towards CO, in its mono-insertion into the Be-H bonds of these species. The small radius of the $Be^{2+}$ ion in conjunction with the non-innocent pendant phosphine moiety of the developed ligands leads to a unique beryllium formyl complex with an ylidic P-$^{CO}$C fragment, whereby the carbon centre, remarkably, datively binds Be. This, alongside reactivity toward carbon dioxide, sheds light on the insertion chemistry of the Be-H bond, complimenting the long-known chemistry of the heavier Alkaline Earth hydrides.

[1] Department of Chemistry, Technische Universität München, Lichtenbergstraße 4, 85747 Garching, Germany. [2] Department of Chemical and Biological Engineering, University of Alabama, Tuscaloosa, AL 35487, USA. ✉email: terrance.hadlington@tum.de

The two naturally occurring neutral carbon oxides, CO and $CO_2$, have long stood as key building blocks in chemical synthesis, most prominently so for CO in the Fischer–Tropsch[1,2], hydroformylation[3,4], and Pauson–Khand processes[5,6]. Whilst the former yields higher hydrocarbons, the latter two involve the incorporation of a single CO molecule, as a formyl group, into unsaturated C–C bonds. Inspired by this, the insertion chemistry of CO into the M–H bond of *f*-block and early *d*-block metals is relatively well studied, though typically leading to C–C coupled products[7–11], with only a small handful of examples leading to stable C1 insertion products[12–14]. The related CO-insertion reaction in main-group hydride systems is extremely rare, the only well-defined examples derived from the group 2 metals, Mg and Ca (Fig. 1). In 2015, both the groups of Jones and Hill showed that [(DippNacnac)Mg-μ2-H]2 (DippNacnac = [(DippNCMe)2CH]−; Dipp = 2,6-iPrC6H3) reacts with CO to yield the ethenediolate moiety through C–C dimerisation of CO, in a similar fashion to *f*-block and early *d*-block metals, whilst it was also shown by Jones et al. that employing the less bulky DepNacnac ligand rather led to CO trimerisation in the formation of a cyclopropanetriolate complex[15,16]. Hill et al. later showed that related Ca chemistry, using [(DippNacnac)Ca-μ2-H]2, also led to ethenediolate complex formation[17]. In contrast to these various CO coupling reactions, the catalytic hydrosilylation of CO was achieved with both Mg and Ca complexes to exclusively yield C1 products[16,17], despite no such products being observed in stoichiometric processes described above.

Related Be chemistry, remarkably, has yet to be explored. Still, one could hypothesise unique reactivity given the high covalent character of Be–X (X = H or C) bonds in combination with the extremely small ionic radius, and thus high charge density, of $Be^{2+}$. Despite interest in the solid-state behaviour of $BeH_2$ over the past decades, even the structure of this binary hydride has long been a point of contention and was only elucidated relatively recently[18,19]. It is perhaps then unsurprising that very little molecular chemistry of beryllium hydrides has been forthcoming, with well-defined examples of such complexes limited to one example of a tris(pyrazolyl)borate beryllium hydride complex, and three examples of NHC-stabilised methyl beryllium hydride complexes[20–23]. The insertion chemistry of the Be–H bond, which is now well established for Mg–Ba, is non-existent outside of isolated examples of heterocyclic ring-opening processes for Be[21,22], which have been subject to computational mechanistic investigations[24]. This is no doubt in part due to the historically renowned toxicity of beryllium, largely in the form of the powdered element[25,26], which in modern labs can be handled quite easily albeit following considerable caution and safety measures. We therefore wished

to develop a reproducible route to soluble beryllium hydride complexes utilising phosphine-functionalised amido pro-ligands, PhiPDippH (PhiPDipp = Ph2PCH2Si(iPr)2N(H)Dipp; Dipp = 2,6-iPr2C6H3), PhPhDippH (PhPhDipp = Ph2PCH2Si(Ph)2N(H)Dipp), and the novel iPPhDippH (iPPhDipp = iPr2PCH2Si(Ph)2N(H)Dipp), developed in our laboratories[27], so as to begin to elucidate the broader reactive behaviour of molecular beryllium hydride complexes, as well as seeking non-innocent involvement of the pendant phosphine arm in our ligand systems. Herein we discuss our initial findings towards this end, in the facile access to dimeric amido-beryllium hydride complexes, and their reactivity towards $CO_2$ and CO. Most interestingly, the reactions of [PhiP-DippBe-μ2-H]2 (6) and [iPPhDippBe-μ2-H]2 (8) with CO proceed to yield the mono-insertion products, with no observed C–C coupling, contrary to previous reports of related Mg and Ca chemistry. This has led to the formation of a stable beryllium formyl complex, 11, being a rare example of such a metal formyl complex derived from a molecular hydride, standing as an exciting start for reactive beryllium hydride chemistry.

## Results and discussion

Our studies began with the synthesis of amido-beryllium halide complexes, 1–3, which was readily achieved by the addition of cooled toluene to a mixture of BeBr2·(Et2O)2 and potassium amides PhiPDippK, PhPhDippK, and iPPhDippK, respectively, at low temperature (Fig. 2)[28]. Slow warming led to essentially quantitative formation of the desired halide complexes as ascertained by $^{31}P\{^{1}H\}$ NMR spectroscopy. Here we note that only compound 1 was isolated, whilst 2 and 3 were generated in situ and used directly following filtration of the crude reaction mixture. We initially sought to access the beryllium hydride complexes via reaction of beryllium alkyl complexes with mild hydride sources, such as boranes and silanes, given the success of this method for Be and heavier group 2 species reported previously[18,20,29]. To this end, 1 was reacted with benzyl potassium in toluene to generate the beryllium benzyl complex 4 (Fig. 2). Upon reaction with an excess of pinacol borane (HBpin), some evidence for the formation of a Be-hydride was found in the HBpin-activation product, 5 (Fig. 2), which is quantitatively formed in this reaction presumably through the further reaction of the target hydride complex 6 (vide infra) with HBpin. Indeed, such ring-opening reactions for HBpin have precedent in alkaline-earth metal chemistry[30,31]. Initial studies regarding the reaction of benzyl complex 4 with the less reactive hydride source, phenyl silane (PhSiH3), did not show any reaction, even after heating for extended periods of time. However, addition of catalytic quantities of potassium *tert*-butoxide (~0.5 mol%) and heating for

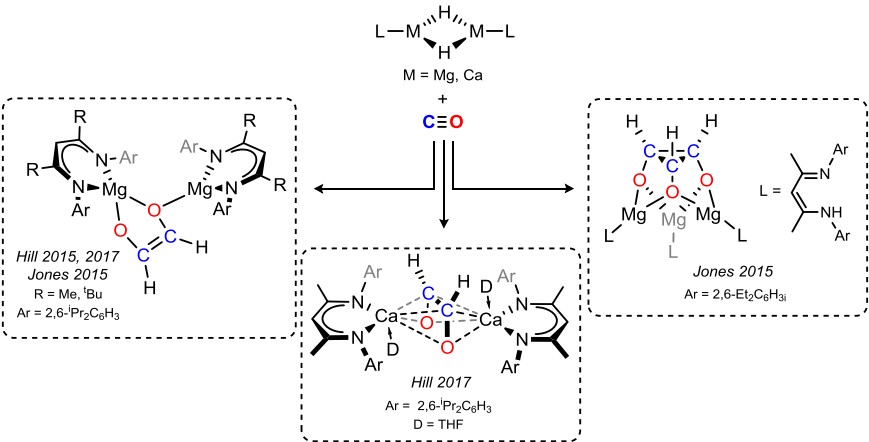

**Fig. 1 CO-dimerisation and -trimerisation through metalation with Alkaline Earth hydride complexes.** Reaction of molecular magnesium and calcium hydride complexes has led to either CO-dimerisation, forming the ethenediolate moiety, or trimerisation, forming the cyclopropanetriolate ligand.

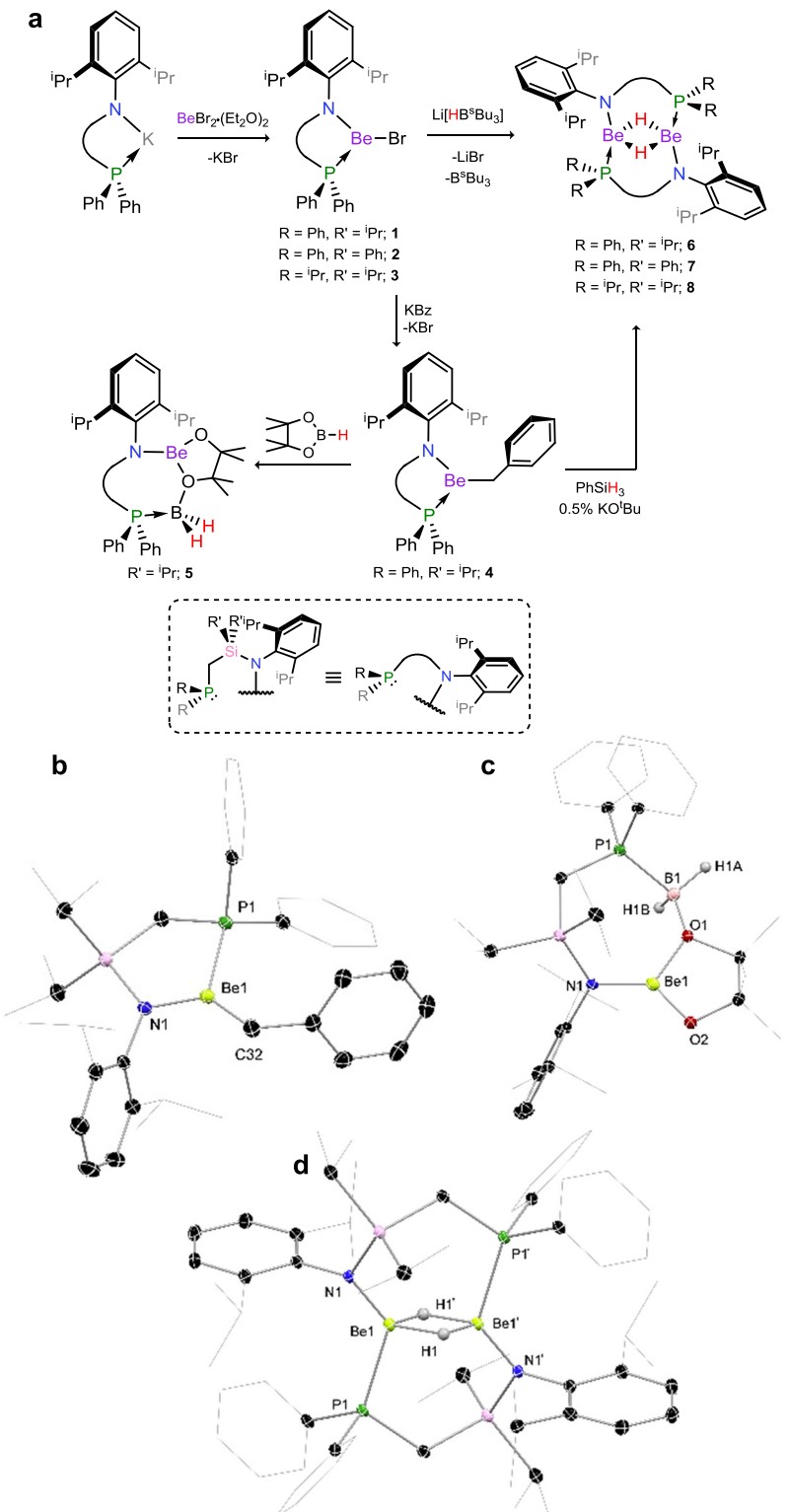

**Fig. 2 Synthesis of compounds 1–8, and the X-ray crystal structures of 4–6.** Beryllium hydride complexes **6**–**8** can be synthesised via addition of a hydrido-borate to amido-beryllium bromides, **1**–**3**, or through addition of PhSiH₃ to the amido-beryllium benzyl complex **4** in the presence of a strong base. Addition of pinacol borane (HBpin) to **4** leads to transient hydride formation followed by HBpin activation in **5**. **a** Synthesis of compounds **1**–**8**. **b** Solid state structure of complex **4**. **c** Solid state structure of complex **5**. **d** Solid state structure of complex **6**. Thermal ellipsoids for **4**–**6** at 30% probability. Selected distances (Å) and angles (°) for **4**: Be1–C32 1.761(4); Be1–N1 1.590(3); Be1–P1 2.217(3); P1–Be1–C32 126.9(1); C32–Be1–N1 132.2(2); N1–Be1–P1 100.8(1). For **5**: Be1–O1 1.633(4); Be1–O2 1.489(3); B1–O1 1.510(4); P1–B1 1.952(3); Be1–N1 1.615(3). For **6**: Be1–P1 2.249(5); Be1–N1 1.623(4); Be1–Be1′ 2.125(6); Be1–H1 1.43(4); Be1–H1′ 1.50(3); P1–Be1–Be1′ 105.1(2); N1–Be1–Be1′ 128.9(3).

24 h led to conversion to one new species, as well as considerable amounts of the silylated ligand, $^{PhiP}$DippN-Si(H)$_2$Ph. Whilst strong bases are known to lead to SiH$_4$ formation through PhSiH$_3$ decomposition, $^1$H NMR spectroscopic analyses of reaction mixtures suggest the presence of PhSi(H)$_2$Bz[32]. Cooling of reaction mixtures leads to the formation of extremely large colourless crystals, up to $10 \times 10 \times 5$ mm, which were ascertained to be the target beryllium hydride complex [$^{PhiP}$DippBe-$\mu_2$-H]$_2$ (6) by X-ray structural analysis (Fig. 2). Despite the monomeric nature of complexes 4 and 5 in the solid state, novel beryllium hydride complex 6 is dimeric, featuring bridging hydride ligands in keeping with the few previously reported beryllium hydride complexes[20]. Each Be centre sits in a 4-coordinate tetrahedral environment, the phosphine arms of the amido ligands now bridging the two beryllium centres, highlighting the flexible coordination of this ligand scaffold. The Be–Be distance ($d_{BeBe} = 2.123(6)$ Å) is in keeping with the two previously reported NHC-stabilised alkyl-beryllium hydride complexes (NHC = N-heterocyclic carbene)[21,22]. Complex 6 is poorly soluble in both apolar and polar organic solvents, and is very sparingly soluble in THF at ambient temperature, but can be dissolved in refluxing THF ($\sim$8 mg mL$^{-1}$). Its $^{31}$P{$^1$H} NMR spectrum shows a major broadened peak at $\delta = -18.6$ ppm, and a second minor peak at $\delta = -22.2$ ppm which we believe to be due to fluctional isomerism in solution (vide infra). Due to solubility issues $^9$Be NMR spectroscopic data could only be recorded at high temperature, where a broad peak centred at $\delta = 5.1$ ppm is observed, with a notable shoulder centred at $\delta = \sim 9$ ppm. We note that, due to the quadrupolar nature of $^9$Be ($s = 3/2$), $^9$Be NMR spectra typically give broad signals, particularly in unsymmetrical binding environments such as those in the presently described compounds. A clear signal for the Be-$H$ fragment could not be observed in the $^1$H NMR spectrum, and indeed all signals are considerably broadened. This broadening, alongside the observation of two signals in the $^{31}$P{$^1$H} and $^9$Be NMR spectra suggests to us isomerisation of 6 in solution. Computationally, we initially investigated a

monomer–dimer equilibrium using the model complex 6' (6' = [$^{PhMe}$XylBe-$\mu_2$-H]$_2$; $^{PhMe}$Xyl = [(Ph$_2$PCH$_2$SiMe$_2$)(Xyl)N]$^-$), which is disfavoured by 24.6 kcal mol$^{-1}$. The non-phosphine-bridged ligand-binding mode for this species, however, sits only 2.2 kcal mol$^{-1}$ higher in energy than the observed solid-state structure (Supplementary Fig. 78), which we believe explains the described observations. X-ray powder diffractograms of powdered crystals of 6 unequivocally show a single structural phase in bulk samples of this crystalline compound (Supplementary Fig. 75), ascertaining its structural 'purity'. Further, addition of 1 equiv. of $^{iPr}$NHC ($^{iPr}$NHC = [(Me)CN($^{iPr}$)]$_2$C:) to suspensions of 6 in D$_8$-THF led to the complete dissolution of the hydride complex and clean formation of a single product, showing resolution of fluctional processes in 6. This new species was found to be the NHC-coordinated monomeric Be–H complex, 9 (Fig. 3), despite the aforementioned energy barrier to monomer formation. Surprisingly, the flanking phosphine arm in 9 does not coordinate Be ($d_{BeP} = 3.579(4)$ Å), likely due to the small ionic radius of this element, leading to, to the best of our knowledge, the first example of a three-coordinate beryllium hydride complex. Crystals of this low-coordinate complex are considerably more air sensitive than four-coordinate 6, bubbling vigorously even under per-fluorinated oil. The $^1$H NMR spectrum of 9 contains a clear broadened multiplet signal at $\delta = 4.21$ ppm integrating to 1H and thus in keeping with the presence of a Be-$H$ moiety. The $^{31}$P{$^1$H} NMR spectrum contains a signal at $\delta = -21.1$ ppm, considerably sharper than that in 6 ($\delta = -18.6$ ppm), in line with the now free phosphine arm, whilst a broad signal is observed in the $^9$Be NMR spectrum at $\delta = 14.9$ ppm, indicative of a three-coordinate Be-centre[33]. Finally, the Be–H stretch is clearly observed in the ATR IR spectrum of 9, with peaks at 1740 and 1800 cm$^{-1}$ (Supplementary Fig. 59). We note that multiple Be–H stretches are observed due to the presence of three independent molecules of 9 in the asymmetric unit of the X-ray crystal structure of this compound.

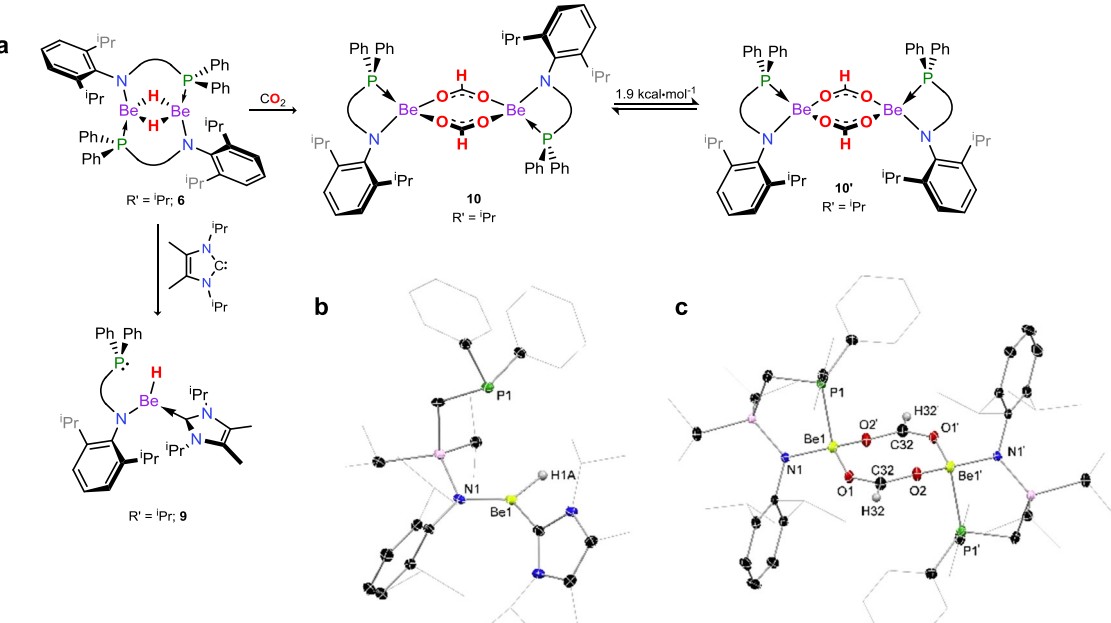

**Fig. 3 Synthesis and X-ray crystal structures of compounds 9 and 10.** Addition of an NHC to the dimeric amido-beryllium hydride complex 6 leads to the 3-coordinate beryllium hydride complex 9, whilst addition of CO$_2$ to 8 leads to insertion into the Be–H bond, forming beryllium formate 10, which shows structural fluctionality in solution. **a** Synthesis of compounds 9 and 10 from dimeric beryllium hydride complex 6. **b** Solid state structure of complex 9. **c** Solid state structure of complex 10. Thermal ellipsoids for 9 and 10 at 30% probability. Selected distance (Å) and angles (°) for 9: Be1–H1A 1.36(3); Be1–C32 1.800(5); Be1–N1 1.616(4); Be1•••P1 3.537(4). For 10: Be1–O1 1.623(3); Be1–O2 1.631(3); O1–C32 1.253(3); O2–C32 1.252(3); Be1–N1 1.652(3); Be1–P1 2.282(3); O1–Be1–O2 113.5(2); O1–Be1–N1 112.5(2); O2–Be1–P1 103.9(2); N1–Be1–P1 97.9(2).

In search for a more expedient, high-yielding route to Be–H complexes, bromide complex **1** was reacted with Li[HB$^S$Bu$_3$] in toluene at low temperature, leading to the rapid formation of copious crystalline material upon warming and standing for 1 h (Fig. 2). This leads to considerably improved reproducible yields of **6** of up to 90%, and can be extended to complexes stabilised by differing phospine-functionalised amido ligands (viz. **2** and **3**, yielding **7** and **8**, respectively). Structurally, all species maintain the same ligand binding motif (Supplementary Fig. 74), bearing bridging phosphine arms and bridging hydride ligands. Compounds **7** and **8**, like **6**, are poorly soluble even in polar organic solvent (~3 mg mL$^{-1}$ in THF at ambient temperature), and show broadened signals in their $^1$H NMR spectra, particularly so for **7**. Nevertheless, with a facile route to amido-beryllium hydride complexes established, we sought to study reactivity towards CO$_2$ and CO, particularly given previously reported reactivity of heavier group 2 hydrides towards these important carbon building blocks[15,17,34,35]. Suspensions of **6** in toluene completely dissolve over the course of 30 min when stirred under an atmosphere of CO$_2$, leading to $^1$H NMR spectra indicating the presence of two species, both of which appear to contain a formate C-*H* moiety, in a 7:3 ratio ($\delta = 7.69$ and 7.83 ppm). Isolation of X-ray quality crystals from these reactions mixtures allowed for structural analysis of the dimeric formate-bridged complex [$^{PhiP}$DippBe-$\mu_2$-{OC(H)O}]$_2$ (**10**, Fig. 3), a binding motif known for molecular main-group formate complexes. Dissolution of the crystals in C$_6$D$_6$ gave $^1$H NMR spectra in keeping with those seen in the crude reaction mixture, with a similar ratio found for the formate signals. The bridging nature of the formate ligands is

further borne out by the observation of triplets for these C-centres in the $^{13}$C NMR spectrum ($\delta = 166.8$ and 172.0, $^3J_{CP} = 3.0$ and 3.1 Hz), due to coupling to both ligand P-centres in this dimer. As for **6**, the clear presence of two sets of signals in NMR spectra for **10** indicated fluctional structural behaviour in solution, despite powdered crystalline samples displaying a single phase in powder X-ray diffraction experiments (Supplementary Fig. 76). In this case we believe isomerism arises from the unsymmetrical nature of the ligand allowing for rotational isomer **10′** (Fig. 3), which is only 1.9 kcal mol$^{-1}$ higher in energy than **10**. As for **6′**, a monomer–dimer equilibrium was found to be unfavourable by 16.1 kcal mol$^{-1}$, whilst the dissociation of the phosphine arms from the Be-centres, yielding three-coordinate Be, is disfavoured by 12.3 kcal mol$^{-1}$ (see Supplementary Fig. 79). Perhaps surprisingly, this reaction represents the first example of the unassisted insertion of CO$_2$ into the M–H bond of a neutral group 2 hydride complex, with the closest examples involving cationic Mg and Ca complexes, and a BCF-coordinated (BCF = (C$_6$F$_5$)$_3$B) Mg and Ca complexes[34].

We then turned our attention to the activation of carbon monoxide. Again, stirring THF-suspensions of **6** under an atmosphere of CO gradually led to the dissolution of this complex, albeit with longer reaction times of 16 h, after which time one major product was observed in the $^{31}$P NMR spectrum of crude reaction mixtures. Large plate-like crystals could be isolated by layering of filtered reaction mixtures with heptane, structural analysis of which revealed a remarkable dimeric beryllium formyl complex [$^{PhiP}$DippBeC(H)O]$_2$ (**11**, Fig. 4a), by insertion of a single equivalent of carbon monoxide into each Be–H bond. **11**

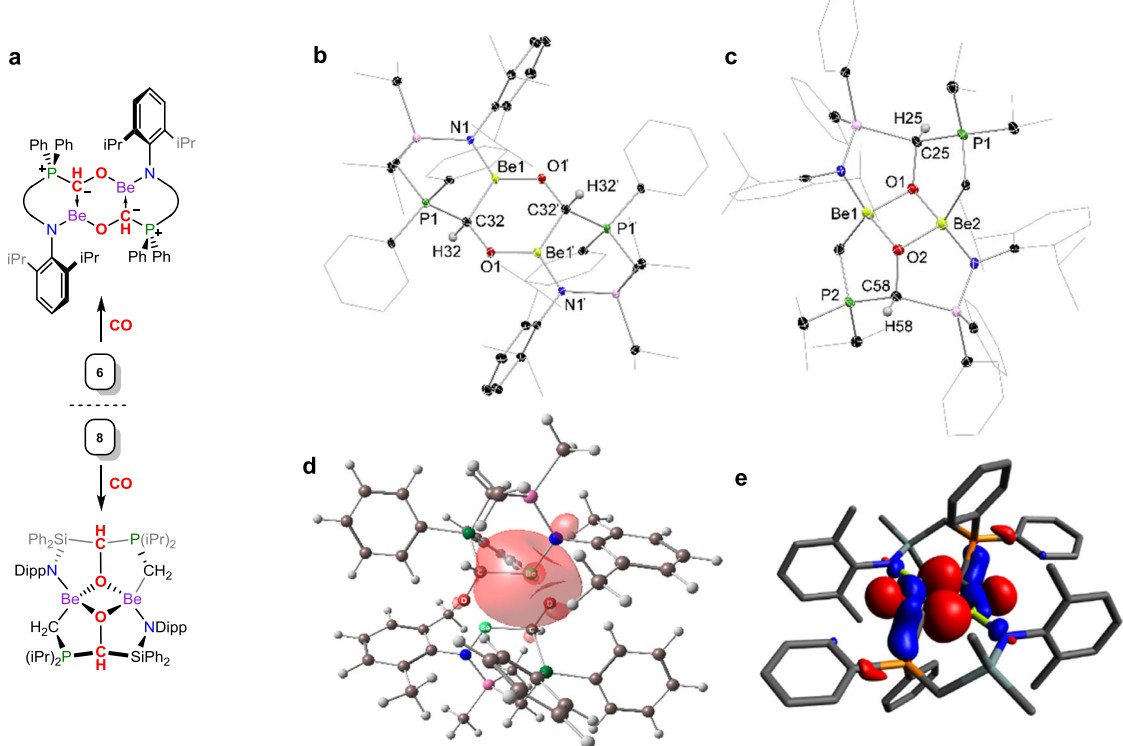

**Fig. 4 Synthesis and X-ray crystal structures of compounds 11 and 12, the DFT-derived HOMO of 11moiety is (−4.04 eV), and the NBO-derived depiction of the dative C → Be interaction in 11moiety is.** The reaction of CO with hydride complex **6** leads insertion into each Be–H bond, leading to **11**. The divergent reaction of CO with hydride complex **8** also proceeds through insertion into the Be–H fragments, but follows through ligand-activation. **a** Synthesis of compounds **6** and **8**. **b** Solid state structure of complex **11**. **c** Solid state structure of complex **12**. Thermal ellipsoids for **11** and **12** at 30% probability. Selected distances (Å) and angles (°) for **11**: Be1–O1′ 1.503(5); Be1–C32 1.807(5); Be1–N1 1.606(4); P1–C32 1.772(3); O1–C32 1.424(3); N1–Be1–C32 120.9(3); C32–Be1–O1′ 119.0(3); N1–Be1–O1′ 120.1(3). For **12**: Be1–O1 1.695(6); Be1–O2 1.691(8); Be1–N1 1.699(8); Be1–C65 1.822(8); O1–C25 1.466(5); P1–C25 1.842(5); P1–C33 1.745(4); O1–Be1–O2 92.7(3); O1–Be1–Be2 86.0(3); O1–Be2–O2 93.6(3); Be1–O2–Be2 87.4(3). **d** The isolated NBO-derived bonding interaction between the C-based lone-pair and Be in **11′**, with an energy of 81.49 kcal mol$^{-1}$. **e** The HOMO of **11′** (−4.04 eV).

contains a planar central $[Be_2C_2O_2]$ ring, supported by two $[^{PhiP}Dipp]$ ligands, the phosphine-arms in which now bind the formyl carbon. The bonding in **11**, whilst unprecedented, deviates from what might be expected for a dimeric formyl complex; the Be–O bonds are very short ($d_{BeO} = 1.503(5)$ Å; $\overline{x}_{Be-O} = 1.620$ Å), and the formyl C–O bonds long ($d_{CO} = 1.424(3)$ Å), pertaining to single bonds. The Be–C bonds, on the other hand, are long, in the order of reported dative C–Be bonds[20–22], with the C–PPh₂ bond ($d_{CP} = 1.772(3)$ Å) in fact slightly shorter than the remaining formal C–P single bonds in this complex. As such, the most fitting bonding model in this species appears to be that given in Fig. 4, with a phosphonium-ylide formed between the CO-carbon and phosphine arm, formal Be–O and C–O single bonds, and C → Be dative bonding. This bonding motif has, to the best of our knowledge, not been described previously, and likely relies in part on the stabilising phosphine arm of the $[^{PhiP}Dipp]$ ligand scaffold. A computational investigation in this bonding situation aids in corroborating the description given above. In the first instance, a natural bond orbital (NBO) analysis of the central $[Be_2C_2O_2]$ moiety of **11′** (Table S3) indicates a high $p$-character lone electron pair on the C atoms, and a high $s$-character vacant orbital on each Be, with an interaction energy of 81.49 kcal mol⁻¹ between these orbitals (Fig. 4c), consistent with the C → Be dative bonding interpretation. Interestingly, this NBO analysis of the Be–O bonds does not indicate a covalent interaction, but rather a lone pair on the O atom, which suggests a highly ionic interaction with the Be centre in line with the large electronegativity difference between these elements, and the calculated very low Wiberg Bond Index of 0.20. Accordingly, the Natural Population Analysis shows +1.63 and −1.04 charge on the Be and O atoms, respectively. However, the weakness of NBO analysis for Be compounds has been noted in previous computational works because it does not consider important 2p atomic orbitals, and only 2 s atomic orbitals, thus overpredicting ionic character of Be bonds[36–38]. We have therefore also calculated the extended Hirshfeld method CM5 charges, the electron and laplacian of electron densities, the kinetic, potential, and total electronic energy densities, and the ellipticity at the Be-related bond critical points (BCP) of the central $Be_2C_2O_2$ moiety in **11'** (see Supplementary Table 4). We found that the CM5 charge shows a considerably more charged balanced, less ionic picture compared to the NPA, having Be and O charges of +0.63 and −0.41, respectively. Topological analysis of the electron density shows positive values for the Laplacian of electron density at the BCP, which is a common error[38], therefore we analysed the more general total electron density descriptor. The total electron density is negative at both the Be–C (−0.016) and Be–O (−0.017) BCPs, which could suggest covalent character for these interactions. Intriguingly, the ellipticity of the Be–C bond at the BCP is 0.12, which may suggest slight double bond character. Overall, this gives a more covalent picture for the Be–O bonding in **11'**, and maintains a dative picture for the Be–C bonding interactions. The relatively low-field shifted signal for the phosphorous centre in the $^{31}P$ NMR spectrum of **11** ($\delta = 10.1$ ppm) is in keeping with phosphonium-character, with the $^{9}Be$ NMR spectrum showing a broad signal at $\delta = 10.0$ ppm. The Be-C(H)O moiety is clearly observable in both the $^{1}H$ ($\delta = 4.86$ ppm; $^{2}J_{HP} = 12.0$ Hz) and $^{13}C\{^1H\}$ ($\delta = 63.3$ ppm; $^{1}J_{CP} = 21.0$ Hz) NMR spectra. Isomerisation to the *syn*-isomer of **11** is observed in solution, which is only 2.0 kcal mol⁻¹ higher in energy than the structurally observed *anti*-isomer (Supplementary Fig. 80 in Supporting Information). Thus, both isomers are observed in NMR spectra of this compound. Examples of such CO mono-insertion products akin to **11** are very rare, and entirely unknown for main-group hydride species. The formation of **11** is likely driven by the very small ionic radius of Be²⁺ and its oxophilicity driving dimerisation, as well as the phosphine arm of the ligand

allowing for the formation of the phosphonium ylide at the core of this compound. In order to expand on the scope of this reaction, hydride complex **8**, featuring the novel $[^{iPPh}Dipp]$ ligand, was reacted with CO. We were surprised to find a rather different major product arising from the reaction, that is penta-cyclic **12** (Fig. 4), formed through cleavage of the ligand's Si–C bond. This species, as **11**, contains an ylidic P–C moiety, giving further evidence for this interaction in stabilising non-CO-coupled products, and highlights the potential diversity of reactive main-group/CO chemistry.

To shed further light on the formation of compound **11**, the mechanism for the formation of this remarkable species was subject to a computational investigation. Calculations based on density functional theory (DFT) employing the model complex **6′** suggest that the mechanism proceeds through two sequential insertions of a CO molecule into the central $[Be_2H_2]$ moiety, each insertion followed by an energetically favourable ligand rearrangement of the phosphine group, which migrates from Be to C. The initial approach of the CO molecules to the Be–H fragments proceeds in a side-on fashion (viz. **TS1**, +16.5 kcal mol⁻¹; Fig. 5), leading to an insertion product which is slightly higher in energy than **6′** (**IM1**, +3.5 kcal mol⁻¹; Fig. 5). P-migration from Be to C (**TS2** and **IM2**; Fig. 5), favoured by 17.3 kcal mol⁻¹, would further suggest that the flanking P-arm plays a key role in stabilising the mono-insertion product, circumventing CO-dimerisation. A similar process then proceeds at the second Be–H moiety, with the overall reaction coordinate being exergonic by 33.4 kcal mol⁻¹, in line with the ready formation of **11**. Reactive intermediates which could lead to structure akin to **12** were not found, however, which insights a more in-depth investigation in the future. It is notable that this mechanism is in stark contrast to that for $[(^{Dipp}Nacnac)Mg$-$\mu_2$-$H]_2$, which proceeds firstly through a monomer–dimer equilibrium of the hydride starting material, followed by the insertion of a single equivalent of CO into both Mg–H bonds, forming $[(^{Dipp}Nacnac)Mg$-$\mu_2$-$(OCH_2)]_2$ as an intermediate, which can then react further with CO[15]. In our system, this form of intermediate is circumvented due to the chelating phosphine arm as well as the small ionic radius of Be²⁺, highlighting a previously unknown pathway for the functionalisation of CO by a metal hydride.

In conclusion, we have developed a facile synthetic method for accessing dimeric beryllium hydride species, which to date are extremely rare and, in our view, understudied. As such, we have investigated their reactivity towards the carbon oxides, CO and CO₂. In the former case this has led to the discovery of a reactive pathway for CO which leads to a unique bonding motif following mono-insertion of CO into each Be–H bond. Overall, this study has shed light on the high reactivity of beryllium hydride species, as well as highlighting the powerful directing effects of the employed phosphine-functionalised amido ligands we are developing in our labs. We now seek to further expand this chemistry, to further define non-innocent capacity of this ligand class.

## Methods

**General considerations**. All experiments and manipulations were carried out under a dry oxygen free argon atmosphere using standard Schlenk techniques, or in an MBraun inert atmosphere glovebox containing an atmosphere of high purity argon. THF and diethyl ether were dried by distillation over a sodium/benzo-phenone mixture and stored over activated 4 Å mol sieves. $C_6D_6$ was dried and stored over a potassium mirror. All other solvents were dried over activated 4 Å mol sieves and degassed prior to use. $^{PhiP}DippK^{27}$, $^{PhPh}DippK^{27}$, $DippN(H)Li^{39}$, $^{i}Pr_2PCH_2Li^{40}$, $BeBr_2·(Et_2O)_2^{28}$, and $BzK^{41}$ were synthesised according to known literature procedures, and are given in the Supplementary Information. All other reagents were used as received. Solution-state NMR spectra were recorded on a Bruker AV 400 or 500 Spectrometer. Solid-state NMR spectra were recorded on a Bruker AV 300 Spectrometer. The $^{1}H$ and $^{13}C\{^1H\}$ NMR spectra were referenced to the residual solvent signals as internal standards. $^{9}Be$, $^{11}B$, $^{29}Si$, and $^{31}P$ NMR spectra were externally calibrated with $BeBr_2·(Et_2O)_2$, $BF_3·OEt_2$, $SiMe_4$, and $H_3PO_4$,

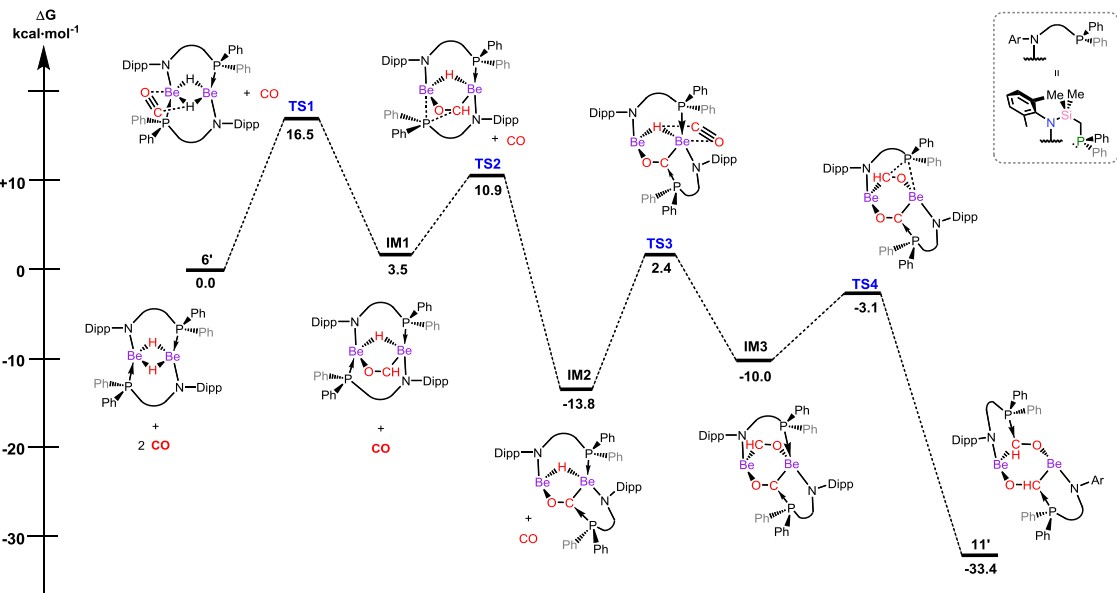

**Fig. 5 Calculated free energy profile for the reaction of 11′ with two equivalents of CO.** The reaction proceeds via two consecutive CO insertion events, each beginning through side-on approach of the CO molecule to the Be–H moiety, and concluded by P-migration from Be to the terminal CO-carbon.

respectively. LIFDI MS spectra were measured at a Waters Micromass LCT TOF mass spectrometer equipped with an LIFDI ion source (LIFDI 700) from Linden CMS GmbH. The samples were dissolved in dry toluene and filtered using a syringe filter under an inert atmosphere. The TOF setup was externally calibrated using polystyrene. ESI–MS was performed on an exactive plus orbitrap spectrometer from Thermo Fischer Scientific. Infra-red spectra were measured with the Alpha FT IR from Bruker containing a platinum diamond ATR device. The compounds were measured as solids under inert conditions in a glovebox. For the ammonia activation experiments water free ammonia 5.0 was used.

*Caution.* Elemental beryllium, most notably as fine powders, and beryllium compounds are regarded as highly toxic and carcinogenic. A severe allergic reaction can also occur if inhaled, with the risk of causing chronic beryllium disease[42]. One should take care and use adequate safety measures (i.e. breathing apparatus, protective clothing, well-ventilated fume-hoods) for any manipulations involving beryllium and compounds containing this element[26].

**Synthesis and characterisation of new compounds. $^{iPPh}$DippK.** A colourless suspension of $^{i}Pr_2PCH_2Li$ (2.00 g, 16.13 mmol) in hexane 50 mL was cooled to −78 °C, and TMEDA added (4.8 mL, 32.26 mmol). The mixture was stirred vigorously and $Ph_2SiCl_2$ (3.4 mL, 16.13 mmol) was added. The mixture was allowed to warm to RT overnight, and all volatiles subsequently removed in vacuo, leaving a colourless oil. Solid DippN(H)Li (2.94 g, 18.6 mmol) was added to the residue, and the flask cooled to −78 °C before the addition of THF (50 mL). The THF was allowed to cool for 5 min, prior to shaking the flask by hand until the majority of the solid had dissolved, allowing free movement of the stir bar. The mixture was then stirred at −78 °C for 1 h, and the cold bath removed to allow the reaction to warm to RT, giving a pale yellow solution. All volatiles were subsequently removed in vacuo and the oily residue extracted with 50 mL hexane, and filtered. The solvent was removed in vacuo and KH (0.84 g, 20.97 mmol) was added. After addition of 50 mL THF, vigorous evolution of gas was observed, and the mixture was stirred for a further 2 h, and then allowed to settle for 16 h. The pale yellow-brown suspension was filtered, and all volatiles were removed in vacuo. To the resulting oil hexane (30 mL) was added, and the mixture heated gently to ~60 °C leading to dissolution of the oil, and formation of small amounts of crystalline solid. All volatiles were removed from the mixture in vacuo, giving a solid residue which was washed twice with hexane (15 mL), and dried in vacuo, yielding $^{iPPh}$DippNK as a free-flowing off-white crystalline powder (6.12 g, 72%).

$^{1}$H NMR (D$_8$-THF, 400 MHz, 298 K): δ = 0.91–1.03 (overlapping d, 24H, $^{i}Pr_2P$-C$H_3$ and Dipp-Pr$^{i}$-C$H_3$), 1.23 (d, 2H, $^2J_{PH}$ = 4.7 Hz, $^{i}Pr_2P$-C$H_2$), 1.48 (sept of d, 2H, $^3J_{HH}$ = 7.2 Hz, $^2J_{PH}$ = 2.5 Hz, $^{i}Pr_2P$-C$H$), 3.96 (sept, 2H, $^3J_{HH}$ = 6.8 Hz, Dipp-Pr$^{i}$-C$H$), 6.21 (t, 1H, $^3J_{HH}$ = 7.4 Hz, Dipp-*p*-C$H$), 6.71 (d, 2H, $^3J_{HH}$ = 7.4 Hz, Dipp-*m*-C$H$), 7.05–7.14 (m, 6H, Ph$_2$Si-Ar-C$H$), 7.57–7.61 (m, 4H, Ph$_2$Si-Ar-C$H$).

$^{13}$C{$^1$H} NMR (D$_8$-THF, 101 MHz, 298 K): δ = 13.2 (d, $^1J_{CP}$ = 31.0 Hz, $^{i}Pr_2P$-C$H_2$-Si), 19.6 (d, $^1J_{CP}$ = 9.6 Hz, $^{i}Pr_2P$-C$H_3$), 21.3 (d, $^1J_{CP}$ = 15.8 Hz, $^{i}Pr_2P$-C$H$), 25.1 (Dipp-Pr$^{i}$-C$H_3$), 27.9 (Dipp-Pr$^{i}$-C$H$), 112.5, 122.9, 127.3, 127.5, 135.8, 141.4, 147.7, 147.8, and 156.5 (Ar-C).

$^{31}$P{$^1$H} NMR (D$_8$-THF, 162 MHz, 298 K): δ = −2.0 (s, CH$_2$-PPh$_2$).

$^{29}$Si{$^1$H} NMR (D$_8$-THF, 99 MHz, 298 K): δ = −47.2c (d, $^2J_{SiP}$ = 11.6 Hz, SiPh$_2$).

**MS/LIFDI-HRMS** found (calcd.) *m/z*: 488.2852 (488.2902) for [M-K]$^+$.

**$^{PhiP}$DippBeBr, 1.** Solid $^{PhiP}$DippK (2 g, 3.8 mmol) and BeBr$_2$·(Et$_2$O)$_2$ (1.2 g, 3.8 mmol) were added to a Schlenk flask, the mixture cooled to −78 °C with rapid stirring, and toluene (40 mL) added. The suspension was stirred at low temperature for 2 h, before being warmed to room temperature, whereby a fine precipitate had formed in place of the initial suspension, with a pale-yellow supernatant solution. To this was added ~10 mL pentane, and the reaction filtered. Removal of all volatiles in vacuo led to a sticky residue, to which was added heptane (10 mL), and the flask stored at −40 °C overnight leading to the formation of microcrystalline solid (1.63 g, 74%), ascertained to be pure 1.

$^{1}$H NMR (C$_6$D$_6$, 400 MHz, 298 K): δ = 0.77 (d, 6H, $^3J_{HH}$ = 7.4 Hz, Si-Pr$^{i}$-C$H_3$), 0.96 (d, 6H, $^3J_{HH}$ = 7.4 Hz, Si-Pr$^{i}$-C$H_3$), 1.20 (m, 2H, Si-Pr$^{i}$-C$H$), 1.28 (d, 6H, $^3J_{HH}$ = 6.8 Hz, Dipp-Pr$^{i}$-C$H_3$), 1.38 (d, 6H, $^3J_{HH}$ = 6.8 Hz, Dipp-Pr$^{i}$-C$H_3$), 1.44 (d, 2H, $^2J_{PH}$ = 13.2 Hz, Ph$_2$P-C$H_2$), 3.53 (sept, 2H, $^3J_{HH}$ = 6.8 Hz, Dipp-Pr$^{i}$-C$H$), 6.92–7.19 (m, 9 H, Ar-C$H$), 7.73–7.79 (m, 4H, Ar-C$H$).

$^{13}$C{$^1$H} NMR (C$_6$D$_6$, 101 MHz, 298 K): δ = 3.1 (d, $^1J_{CP}$ = 6.4 Hz, Ph$_2$P-C$H_2$), 13.9 and 14.0 (Si-iPr-CH), 18.2 and 18.8 (Si-iPr-C$H_3$), 23.6 (Dipp-Pr$^{i}$-CH), 26.4 and 28 .7 (Dipp-Pr$^{i}$-C$H_3$), 123.6, 123.7, 129.5 (d), 131.3 (m), 131.6, 132.6, 144.6, and 145.0 (d) (Ar-C).

$^{31}$P{$^1$H} NMR (C$_6$D$_6$, 162 MHz, 298 K): δ = −26.1 (br s, Ph$_2$P-Be).

$^{29}$Si{$^1$H} NMR (C$_6$D$_6$, 99 MHz, 298 K): δ = 9.6 (d, $^2J_{SiP}$ = 8.6 Hz, SiiPr$_2$).

$^{9}$Be NMR (C$_6$D$_6$, 58 MHz, 298 K), δ = 13.7.

**MS/LIFDI-HRMS** found (calcd.) *m/z*: 578.2155 (578.2182) for [M]$^+$.

**$^{PhiP}$DippBeBz, 4.** Benzyl potassium (290 mg, 2.25 mmol) and 1 (1 g, 1.73 mmol) were added to a Schlenk flask, and toluene (15 mL) added at ambient temperature. The resulting red suspension was placed in an ultrasonic bath for 15 min, at which time only a small amount of red solid remained. The reaction mixture was filtered, concentrated to 2 mL, and layered with heptane (7 mL), leading to the formation of a large amount of X-ray quality colourless crystals of 4 over the course of 24 h (630 mg, 62%).

$^{1}$H NMR (C$_6$D$_6$, 400 MHz, 298 K): δ = 0.77 (d, 6H, $^3J_{HH}$ = 7.4 Hz, Si-Pr$^{i}$-C$H_3$), 0.98 (d, 6H, $^3J_{HH}$ = 7.4 Hz, Si-Pr$^{i}$-C$H_3$), 1.21 (m, 2H, Si-Pr$^{i}$-C$H$), 1.24 (d, 6H, $^3J_{HH}$ = 6.8 Hz, Dipp-Pr$^{i}$-C$H_3$), 1.38 (d, 6H, $^3J_{HH}$ = 6.8 Hz, Dipp-Pr$^{i}$-C$H_3$), 1.38 (d, 2H, $^2J_{PH}$ = 12.4 Hz, Ph$_2$P-C$H_2$), 1.95 (d, 2H, $^3J_{PH}$ = 4.5 Hz, Be-C$H_2$-Ph), 3.62 (sept, 2H, $^3J_{HH}$ = 6.8 Hz, Dipp-Pr$^{i}$-C$H$), 6.87 (m, 1H, Ar-C$H$), 6.97–7.03 (m, 6 H, Ar-C$H$), 7.05–7.20 (m, 11H, Ar-C$H$).

$^{13}$C{$^1$H} NMR (THF-d$_8$, 101 MHz, 298 K): δ = 4.7 (d, $^1J_{CP}$ = 2.0 Hz, Ph$_2$P-C$H_2$), 14.0 (Si-iPr-CH), 18.4 and 19.0 (Si-iPr-C$H_3$), 23.9 and 26.2 (Dipp-Pr$^{i}$-C$H_3$), 24.0 (br, Be-C$H_2$Ph), 28.4 (Dipp-Pr$^{i}$-CH), 121.0, 123.2, 123.6, 125.7, 127.5, 128.6, 128.8, 129.1 (d), 129.3, 130.6, 132.5, 132.8, 133.0 (d), 144.6, 146.5 (d), and 151.5 (Ar-C).

$^{9}$Be NMR (C$_6$D$_6$, 58 MHz, 298 K), δ = 17.0.

$^{29}$Si{$^1$H} NMR (C$_6$D$_6$, 99 MHz, 298 K): δ = 8.5 (d, $^2J_{SiP}$ = 10.2 Hz, Si-iPr$_2$).

**$^{31}$P{$^{1}$H} NMR** (C$_6$D$_6$, 162 MHz, 298 K): $\delta = -19.5$ (br s, Ph$_2$P-Be).
**MS/LIFDI-HRMS** found (calcd.) $m/z$: 497.2980 (497.3019) for [M-C$_7$H$_7$]$^+$.

**$^{PhiP}$DippBe(pin)BH$_2$, 5.** To a toluene solution of **4** (200 mg, 0.20 mmol) in toluene (5 mL) was added neat HBpin (60 μL, 0.60 mmol). The reaction mixture was shaken, and allowed to stand for 30 min, before all volatiles were removed in vacuo. The oily residue was extracted in Et$_2$O (10 mL), filtered, and concentrated to ~3 mL. Storage of this solution at ambient temperature overnight led to the formation of a large crop of colourless crystals of **5** (155 mg, 62%), suitable for X-ray crystallographic analysis.

**$^{1}$H NMR** (C$_6$D$_6$, 400 MHz, 298 K): $\delta = 0.94$ (d, 6H, $^3J_{HH} = 7.4$ Hz, Si-Pr$^i$-C$H_3$), 1.03 (d, 6H, $^3J_{HH} = 7.4$ Hz, Si-Pr$^i$-C$H_3$), 1.14 (s, 6H, pin-$Me$), 1.21 (sept, 2H, $^3J_{HH} = 7.4$ Hz, Si-Pr$^i$-C$H$), 1.36 (d, 12H, $^3J_{HH} = 6.8$ Hz, Dipp-Pr$^i$-C$H_3$), 1.43 (s, 6H, pin-$Me$), 1.79 (d, 2H, $^2J_{PH} = 17.0$ Hz, Ph$_2$P-C$H_2$), 3.51 (v br, 2H, B$H_2$), 3.73 (sept, 2H, $^3J_{HH} = 6.8$ Hz, Dipp-Pr$^i$-C$H$), 7.00–7.14 (m, 9 H, Ar-C$H$), 7.50–7.58 (m, 4H, Ar-C$H$).
**$^{13}$C{$^{1}$H} NMR** (C$_6$D$_6$, 101 MHz, 298 K): $\delta = 6.2$ (d, $^1J_{CP} = 17.3$ Hz, Ph$_2$P-C$H_2$), 15.2 and 15.3 (Si-$^i$Pr-CH), 19.2 and 19.6 (Si-$^i$Pr-CH$_3$), 22.7 and 27.5 (pin-CH$_3$), 23.7 (Dipp-Pr$^i$-CH$_3$), 26.4 and 28.3 (Dipp-Pr$^i$-CH$_3$), 92.3 and 92.4 (pin-$C$(Me)$_2$), 123.0, 123.4, 129.1 (d), 129.8, 130.4, 131.6, 133.4 (d), 146.2, and 146.3 (Ar-$C$);
**$^{9}$Be NMR** (C$_6$D$_6$, 58 MHz, 298 K), $\delta = 8.9$.
**$^{11}$B{$^{1}$H} NMR** (C$_6$D$_6$, 43 MHz, 298 K): $\delta = -12.1$ (br, Be(pin)B$H_2$).
**$^{29}$Si{$^{1}$H} NMR** (C$_6$D$_6$, 99 MHz, 298 K): $\delta = -3.0$.
**$^{31}$P{$^{1}$H} NMR** (C$_6$D$_6$, 162 MHz, 298 K): $\delta = -0.9$ (s, Ph$_2$P-Be).
**IR**, $\nu$ cm$^{-1}$ (ATR): 2340, 2385, 2440 (B-H).
**MS/LIFDI-HRMS** found (calcd.) $m/z$: 626.4085 (626.4111) for [M]$^+$.

**[$^{PhiP}$DippBe-μ$_2$-H]$_2$, 6.** Method (a): To a toluene (20 mL) solution of **4** (500 mg, 0.85 mmol) was added KO$^t$Bu (~1 mg) and PhSiH$_3$ (0.52 mL, 4.25 mmol). The pale-yellow solution was heated to 65 °C for 18 h, at which stage $^{31}$P NMR spectroscopic analysis indicated that all starting material had been consumed. Upon cooling the reaction mixture to ambient temperature large colourless crystals of hydride complex **6** formed (125 mg, 30%), which were suitable for X-ray crystallographic analysis.

Method (b): A solution of **1** (1 g, 1.73 mmol) in tolene (40 mL) was cooled, without stirring, to $-78$ °C. A THF solution of L-selectride (2.08 mL, 1 M) was then rapidly added, and the solution shaken whilst cold for a few seconds. The cold bath was then removed, allowed the reaction to reach room temperature. After standing for 1 h copious amounts of colourless crystalline solid had formed. Decanting the solution, washing with pentane (10 mL), and drying under vacuum afforded **6** (790 mg, 92%) as an analytically pure crystalline solid.

**$^{1}$H NMR** (D$_8$-THF, 400 MHz, 338 K): $\delta = 0.50$–$0.75$ (br, 9H, Dipp-Pr$^i$-C$H_3$ and Si-Pr$^i$-C$H_3$), 0.77–1.48 (br, 17H, Dipp-Pr$^i$-C$H_3$, Si-Pr$^i$-C$H_3$, and Si-Pr$^i$-C$H$), 1.67 (br, 2H, Ph$_2$P-C$H_2$), 3.81 (br, 3H, Be-$H$, and Dipp-Pr$^i$-C$H$), 6.86 (br, 3H, Ar-C$H$), 7.12–7.32 (br m, 6 H, Ar-C$H$), 7.36–7.70 (br m, 4H, Ar-C$H$).
**$^{9}$Be NMR** (D$_8$-THF, 58 MHz, 333 K), $\delta = 5.1$ ([$^{PhiP}$DippBeH]$_2$), and 9.4 ($^{PhiP}$DippBeH).
**$^{31}$P{$^{1}$H} NMR** (D$_8$-THF, 162 MHz, 333 K): $\delta = -17.8$ (br s, [$^{PhiP}$DippBeH]$_2$, Ph$_2$P–Be), and $-22.2$ (br s, $^{PhiP}$DippBeH, Ph$_2$P–Be).
**MS/LIFDI-HRMS** found (calcd.) $m/z$: 996.6086 (996.6199) for [M]$^+$.

N.B. The poor solubility as well the broadening of signals due to fluxional solution processes of **6**, even at high temperatures, precluded the collection of meaningful $^{13}$C{$^{1}$H} and $^{29}$Si{$^{1}$H} NMR spectra of this species.

**[$^{PhPh}$DippBe-μ$_2$-H]$_2$, 7.** Solid $^{PhPh}$DippNK (1 g, 1.68 mmol) and BeBr$_2$·(Et$_2$O)$_2$ (530 mg, 1.68 mmol) were added to a Schlenk flask, the mixture cooled to $-78$ °C with rapid stirring, and toluene (25 mL) added. The suspension was stirred at low temperature for 2 h, before being warmed to room temperature, whereby a fine precipitate had formed in place of the initial suspension, with a pale yellow supernatant solution. To this was added ~10 mL pentane, and the reaction filtered. The filtrate was cooled to $-78$ °C, without stirring, and a THF solution of L-Selectride (2.02 mL, 1 M) was added. The mixture was briefly shaken, the cold bath removed, and the reaction warmed to ambient temperature. After standing for 1 h a large crop of colourless crystals had formed, which were isolated by filtration, washed with pentane (5 mL), and dried under vacuum to afford **7** (810 mg, 85%).
**$^{1}$H NMR** (D$_8$-THF, 400 MHz, 298 K): $\delta = 0.34$ (v br, 12H, Dipp-Pr$^i$-C$H_3$), 0.54 (v br, 9H, Dipp-Pr$^i$-C$H_3$), 0.87 (br m, 3H, Dipp-Pr$^i$-C$H_3$), 1.29 (m, 2H, Ph$_2$P-C$H_2$), 2.13 (m, 2H, $^3J_{HH} = 6.8$ Hz, Ph$_2$P-C$H_2$), 3.84 (m, 2H and 2H, Dipp-Pr$^i$-C$H$ and Be-$H$), 4.17 (br, 2H, Dipp-Pr$^i$-C$H$), 6.85-7.47 (v br, 46H, Ar-C$H$).
**$^{9}$Be NMR** (D$_8$-THF, 58 MHz, 298 K), $\delta = 5.3$.
**$^{31}$P{$^{1}$H} NMR** (D$_8$-THF, 162 MHz, 298 K): $\delta = -16.4$ (br s, Ph$_2$P-Be).
**MS/LIFDI-HRMS** found (calcd.) $m/z$: 1132.5518 (1132.5573) for [M]$^+$.

N.B. The poor solubility as well the broadening of signals in the NMR spectra of **7** precluded the acquisition of meaningful $^{13}$C{$^{1}$H} and $^{29}$Si{$^{1}$H} spectra for this species.

**[$^{iPPh}$DippBe-μ$_2$-H]$_2$, 8.** The procedure for the synthesis of **7** was used, with $^{iPPh}$DippNK (500 mg, 0.95 mmol), BeBr$_2$·(Et$_2$O)$_2$ (300 mg, 0.95 mmol), and L-selectride (1.14 mL, 1 M in THF). Compound **8** was isolated as large colourless crystals suitable for X-ray crystallographic analysis (370 mg, 78%).
**$^{1}$H NMR** (D$_8$-THF, 400 MHz, 338 K): $\delta = -0.02$ (br m, 6H, P-Pr$^i$-C$H_3$), 0.46 (br m, 6H, P-Pr$^i$-C$H_3$), 0.49 (d, 6H, $^3J_{HH} = 6.8$ Hz, Dipp-Pr$^i$-C$H_3$), 0.69 (d, 6H,

$^3J_{HH} = 6.8$ Hz, Dipp-Pr$^i$-C$H_3$), 0.86 (m, 2H, P-Pr$^i$-C$H$), 1.07 (br m, 6H, P-Pr$^i$-C$H_3$), 1.19 (m, 12H, Dipp-Pr$^i$-C$H_3$), 1.53 (br m, 6H and 4H, P-Pr$^i$-C$H_3$ and iPr$_2$-C$H_2$), 2.17 (br sept, 2H, Dipp-Pr$^i$-C$H$), 3.69 (br m, 2H and 2H, P-Pr$^i$-C$H$ and Be-$H$), 4.08 (sept, 2H, $^3J_{HH} = 6.8$ Hz, Dipp-Pr$^i$-C$H$), 6.87 (m, 4H, Ar-C$H$), 6.91–6.97 (m, 10 H, Ar-C$H$), 7.05–7.20 (m, 8H, Ar-C$H$), 7.90 (m, 4H, Ar-C$H$).
**$^{13}$C{$^{1}$H} NMR** (D$_8$-THF, 101 MHz, 298 K): $\delta = 4.9$, 17.1, 17.5 and 18.2 (P-Pr$^i$-C$H_3$), 21.7 and 23.9 (m, P-Pr$^i$-CH), 20.3, 24.5, 26.2 and 27.5 (Dipp-$^i$Pr-CH$_3$), 28.5 and 28.6 (Dipp-$^i$Pr-CH), 123.1, 124.1, 125.1, 127.7 (d), 129.2 (d), 136.5, 137.2, 142.2, 146.2, 148.1, and 152.5 (Ar-C).
**$^{9}$Be NMR** (D$_8$-THF, 58 MHz, 338 K), $\delta = 2.8$.
**$^{31}$P{$^{1}$H} NMR** (D$_8$-THF, 162 MHz, 338 K): $\delta = -6.9$ (br s, $^i$Pr$_2$P–Be).
**MS/LIFDI-HRMS** found (calcd.) $m/z$: 996.6129 (996.6199) for [M]$^+$.

**$^{PhiP}$DippBeH·$^{iPr}$NHC, 9.** Solid **6** (150 mg, 0.30 mmol) and $^{iPr}$NHC (55 mg, 0.30 mmol) were mixed in a Schlenk flask, and toluene (5 mL) added. Stirring the suspension for 5 min led to dissolution of the suspension, at which point all volatiles were remove in vacuo. The residue was extracted in pentane (10 mL), filtered, and the resulting solution concentrated to 2 mL. Storage at $-40$ °C for 1 week led to the formation of colourless crystals of **9** (85 mg, 43%), suitable for X-ray crystallographic analysis.
**$^{1}$H NMR** (C$_6$D$_6$, 400 MHz, 298 K): $\delta = 0.73$ (d, 6H, $^3J_{HH} = 6.8$ Hz, Dipp-Pr$^i$-C$H_3$), 0.83 (d, 6H, $^3J_{HH} = 7.4$ Hz, Si-Pr$^i$-C$H_3$), 0.94 (d, 6H, $^3J_{HH} = 7.4$ Hz, Si-Pr$^i$-C$H_3$), 1.14 (d, 6H, $^3J_{HH} = 6.8$ Hz, Dipp-Pr$^i$-C$H_3$), 1.15 (d, 12H, $^3J_{HH} = 6.9$ Hz, NHC-$^i$Pr-CH$_3$) 1.28 (sept, 2H, $^3J_{HH} = 7.4$ Hz, Si-Pr$^i$-C$H$), 1.66 (d, 2H, $^2J_{PH} = 4.8$ Hz, Ph$_2$P-C$H_2$), 2.15 (s, 6H, NHC-$Me$), 3.82 (sept, 2H, $^3J_{HH} = 6.8$ Hz, Dipp-Pr$^i$-C$H$), 4.17 (br m, 1H, Be-$H$), 4.80 (sept, 2H, $^3J_{HH} = 6.9$ Hz, NHC-$^i$Pr-C$H$), 6.79 (t, 1H, Dipp-$p$-C$H$), 6.94 (d, 2H, Dipp-$m$-C$H$), 7.12–7.23 (m, 6 H, Ar-C$H$), 7.46–7.52 (m, 4H, Ar-C$H$).
**$^{13}$C{$^{1}$H} NMR** (C$_6$D$_6$, 101 MHz, 298 K): $\delta = 9.9$ (NHC-NC$Me$), 13.2 (d, $^1J_{CP} = 32.5$ Hz, Ph$_2$P-C$H_2$), 16.2 (d, $^3J_{CP} = 2.5$ Hz, Si-$^i$Pr-CH), 20.4 and 20.5 (Dipp-$^i$Pr-CH$_3$), 21.5 (Si-$^i$Pr-CH$_3$), 25.3 (NHC-$^i$Pr-CH$_3$), 27.8 (Dipp-$^i$Pr-CH), 52.7 (NHC-$^i$Pr-CH), 121.9 (NHC-NC$Me$), 124.2, 125.2, 125.7, 128.6, 129.3, 133.6 (d), 144.4 (d), 144.7, 151.9 (Ar-C), and 173.1 (NHC-$C$:).
**$^{31}$P{$^{1}$H} NMR** (C$_6$D$_6$, 162 MHz, 298 K): $\delta = -21.1$ (s, $^i$Pr$_2$P-Be).
**$^{29}$Si{$^{1}$H} NMR** (C$_6$D$_6$, 99 MHz, 298 K): $\delta = -0.1$ (d, $^2J_{SiP} = 15.5$ Hz, Si-$^i$Pr$_2$).
**$^{9}$Be NMR** (C$_6$D$_6$, 58 MHz, 298 K), $\delta = 14.9$.
**MS/LIFDI-HRMS** found (calcd.) $m/z$: 678.4589 (678.4723) for [M]$^+$.
**IR**, $\nu$ cm$^{-1}$ (ATR): 1742 (m, Be-$H$), 1800 (w, Be-$H$).

**[$^{PhiP}$DippBeCO(H)O]$_2$, 10.** Compound **8** (200 mg, 0.20 mmol) was suspended in toluene (5 mL), and the Schlenk flask purged with dry CO$_2$ through a septum, using a needle adapter. The flask was quickly sealed, and stirred for 16 h, whereupon a clear, colourless solution had formed. Removal of the stir bar, concentration to ~1 mL, and layering with hexane (5 mL) led to the formation of a large crop of colourless crystals of **10** (135 mg, 63%).
**$^{1}$H NMR** (C$_6$D$_6$, 400 MHz, 298 K): $\delta = 0.81$ (d, 6H (conf. (a)), $^3J_{HH} = 6.8$ Hz, Dipp-Pr$^i$-C$H_3$), 0.95 (overlapping d, 6H (conf. (a)) and 12H (conf. (b)), Dipp-Pr$^i$-C$H_3$ and Si-Pr$^i$-C$H_3$), 1.19 (overlapping d, 12H (conf. (a)) and 9H (conf. (b)), Dipp-Pr$^i$-C$H_3$ and Si-Pr$^i$-C$H_3$), 1.24 (d, 3H (conf. (b)), $^3J_{HH} = 6.8$ Hz, Dipp-Pr$^i$-C$H_3$), 1.16-1-31 (2H (conf. (a)) and 2H (conf. (b)), Si-Pr$^i$-C$H$), 1.52 (d, 2H (conf. (b)), $^2J_{PH} = 10.9$ Hz, Ph$_2$P-C$H_2$), 1.52 (d, 2H (conf. (a)), $^2J_{PH} = 9.2$ Hz, Ph$_2$P-C$H_2$), 3.67 (sept, 2H (conf. (a)), $^3J_{HH} = 6.8$ Hz, Dipp-Pr$^i$-C$H$), 3.72 (sept, 2H (conf. (b)), $^3J_{HH} = 6.8$ Hz, Dipp-Pr$^i$-C$H$), 7.06 (overlapping m, 9H (conf. (a)) and 9H (conf. (b)), Ar-C$H$), 7.38 (m, 4H (conf. (b)), Ar-C$H$), 7.52 (m, 4H (conf. (b)), Ar-C$H$), 7.05–7.20 (m, 11H, Ar-C$H$), 7.69 (s, 1H ((conf. (b)), Be-O$C$(H)O–Be), 7.83 (s, 1H (conf. (a)), Be-O$C$(H)O-Be).
**$^{13}$C{$^{1}$H} NMR** (C$_6$D$_6$, 101 MHz, 298 K): $\delta = 6.3$ and 6.5 (2× d, $^1J_{CP} = 8.6$ and 4.2 Hz, Ph$_2$P-C$H_2$), 15.1 and 15.4 (2× d, $^3J_{CP} = 1.9$ and 2.5 Hz, Si-$^i$Pr-CH), 19.0, 19.1, 19.6, and 19.8 (Si-$^i$Pr-CH$_3$), 23.9, 24.0, 24.7, and 26.8 (Dipp-Pr$^i$-CH$_3$), 27.5 and 27.6 (Dipp-Pr$^i$-CH), 121.9, 122.5, 123.5, 123.8, 128.9 and 129.0 (overlapping d), 129.8, 132.7 (d), 132.9 (d), 135.2 (d), 136.2 (d), 146.2, 146.3, 147.8 (d), and 148.3(Ar-C), 166.8 and 172.0 (2× t, $^3J_{CP} = 3.0$ and 3.1 Hz, Be-O$C$(H)O–Be).
**$^{31}$P{$^{1}$H} NMR** (C$_6$D$_6$, 162 MHz, 298 K): $\delta = -22.3$ (s, Ph$_2$P–BeOC(H)O, conf. (b)), and $-23.9$ (s, Ph$_2$P–BeOC(H)O, conf. (a)).
**$^{29}$Si{$^{1}$H} NMR** (C$_6$D$_6$, 99 MHz, 298 K): $\delta = 0.0$ (d, $^2J_{SiP} = 4.5$ Hz, iPr$_2$Si-CH$_2$-PPh$_2$, conf. (a)), 0.13 (d, $^2J_{SiP} = 4.7$ Hz, iPr$_2$Si-CH$_2$-PPh$_2$, conf. (a)).
**$^{9}$Be NMR** (C$_6$D$_6$, 58 MHz, 298 K), $\delta = 3.5$ (v br).
**IR**, $\nu$ cm$^{-1}$ (ATR): 1650 (s, Be–$O$ = $C$(H)O)
**MS/LIFDI-HRMS** found (calcd.) $m/z$: 1084.6012 (1084.6001) for [M$^+$].

**[$^{PhiP}$DippBeC(H)O]$_2$, 11.** Compound **6** (300 mg, 0.30 mmol) was suspended in toluene (6 mL), and the flask subjected to three freeze–pump–thaw cycles, backfilling with CO each time. The flask was subsequently sealed and stirred at ambient temperature for 2 days, at which stage a clear, colourless solution had formed. The reaction mixture was then filtered and layered with heptane (15 mL), leading to the formation of X-ray quality colourless crystals of **11** (105 mg, 33%).
**$^{1}$H NMR** (D$_8$-THF, 400 MHz, 338 K): $\delta = 0.49$ (m, 12H, Si-Pr$^i$-C$H_3$), 0.78 (overlapping d/sept, for d: $^3J_{HH} = 7.4$ Hz, 10H, Si-Pr$^i$-C$H_3$ and Si-Pr$^i$-C$H$), 0.95 (overlapping d, 18H, Si-Pr$^i$-C$H_3$ and Dipp-Pr$^i$-C$H_3$), 1.10 (d, 6H, $^3J_{HH} = 6.8$ Hz, Dipp-Pr$^i$-C$H_3$), 1.38 (br, 6H, Dipp-Pr$^i$-C$H_3$), 1.46 (m, 2H, Ph$_2$P-C$H_2$), 1.96 (d, 2H,

$Ph_2P$-$CH_2$), 2.86 (sept, 2H, $^3J_{HH} = 6.8$ Hz, Dipp-$Pr^i$-$CH$), 3.56 (br sept, 2H, Dipp-$Pr^i$-$CH$), 4.86 (d, 2H, $^2J_{PH} = 12.0$ Hz, Be-$C(H)O$), 6.80–6.92 (m, 6 H, Ar-$CH$), 7.06–7.14 (m, 3H, Ar-$CH$), 7.28–7.36 (m, 3H, Ar-$CH$), 7.49–7.60 (m, 6H, Ar-$CH$), 7.94–8.0 (m, 4H, Ar-$CH$), 8.20–8.30 (m, 3H, Ar-$CH$).

$^{13}C\{^1H\}$ **NMR** ($D_8$-THF, 101 MHz, 298 K): $\delta = 2.4$ (d, $^1J_{CP} = 44.0$ Hz, $Ph_2P$-$CH_2$), 15.8 and 16.2 (Si-$^iPr$-CH), 18.7, 19.4, 19.7, and 20.3 (Si-$^iPr$-$CH_3$), 24.0, and 24.1 (Si-$^iPr$-$CH_3$), 26.0, 26.88, 27.9, and 28.6 (Dipp-$Pr^i$-$CH_3$), 63.3 (Be-$C$(H)O, $^1J_{CP} = 21.0$ Hz), 121.6, 122.8, 123.46, 129.2, 129.3, 129.4, 129.9 (d), 131.9, 132.4, 132.9 (d), 133.5, 133.6, 145.5, 146.5, and 148.8 (Ar-$C$).

$^{31}P\{^1H\}$ **NMR** ($D_8$-THF, 162 MHz, 298 K): $\delta = 9.5$ (br s, $Ph_2P$-Be).

$^9$**Be NMR** ($D_8$-THF, 58 MHz, 298 K), $\delta = 10.0$.

**MS/LIFDI-HRMS** found (calcd.) $m/z$: 1052.6149 (1052.6103) for $[M]^+$.

N.B. Presumably due to broadening alongside the low solubility of **11**, meaningful $^{29}$Si NMR spectra could not be acquired.

**[$^{iPPh}$DippBeC(H)O]$_2$, 12.** Compound **8** (150 mg, 0.15 mmol) was dissolved in THF (5 mL), and the flask subjected to three freeze–pump–thaw cycles, backfilling with CO each time. The reaction mixture was then stirred at ambient temperature for 2 days, and all volatiles subsequently removed in vacuo. The residue was redissolved in $Et_2O$, filtered, and kept at ambient temperature for 3 days, leading to the formation of large colourless block-shaped crystals of **12** (65 mg, 41%), suitable for X-ray crystallographic analysis.

$^1$**H NMR** ($D_8$-THF, 400 MHz, 298 K): $\delta = -0.32$ (m, 2H, $^iPr_2P$-$CH_2$-Be), 0.02 (m, 2H, $^iPr_2P$-$CH_2$-Be), 0.26 (d, 6H, $^3J_{HH} = 6.8$ Hz, Dipp-$Pr^i$-$CH_3$), 0.52 (d, 3H, $^3J_{HH} = 7.2$ Hz, P-$Pr^i$-$CH_3$), 0.55 (d, 3H, $^3J_{HH} = 7.2$ Hz, P-$Pr^i$-$CH_3$), 0.68 (d, 3H, $^3J_{HH} = 7.2$ Hz, P-$Pr^i$-$CH_3$), 0.72 (d, 3H, $^3J_{HH} = 7.2$ Hz, P-$Pr^i$-$CH_3$), 0.81–0.89 (overlapping d, 12H, P-$Pr^i$-$CH_3$ and Dipp-$Pr^i$-$CH_3$), 0.93 (d, 6H, $^3J_{HH} = 6.8$ Hz, Dipp-$Pr^i$-$CH_3$), 1.04 (d, 3H, $^3J_{HH} = 7.2$ Hz, P-$Pr^i$-$CH_3$), 1.07 (d, 3H, $^3J_{HH} = 7.2$ Hz, P-$Pr^i$-$CH_3$), 1.42 (sept, 2H, $^3J_{HH} = 7.2$ Hz, P-$Pr^i$-$CH$), 1.75 (overlapping sept, 2H, P-$Pr^i$-$CH$), 3.31 (sept, 2H, $^3J_{HH} = 6.8$ Hz, Dipp-$Pr^i$-$CH$), 4.20 (sept, 2H, $^3J_{HH} = 6.8$ Hz, Dipp-$Pr^i$-$CH$), 4.77 (d, 2H, $^2J_{PH} = 8.4$ Hz, BeOC(H)P), 6.72 (m, 2H, Ar-$CH$), 6.80 (m, 2H, Ar-$CH$), 6.97 (m, 2H, Ar-$CH$), 7.27 (m, 10H, Ar-$CH$), 7.31 (m, 2H, Ar-$CH$), 7.74 (m, 8H, Ar-$CH$).

$^{13}C\{^1H\}$ **NMR** ($D_8$-THF, 101 MHz, 298 K): $\delta = 17.5$ (d, $^1J_{CP} = 15.0$ Hz, $^iPr_2P$-$CH_2$), 18.7 and 19.6 ($^iPr_2P$-$^iPr$-$CH_3$), 24.5 and 24.9 (Dipp-$^iPr$-$CH_3$), 26.8 and 27.7 (Dipp-$^iPr$-CH), 28.3 and 28.5 ($^iPr_2P$-$^iPr$-CH), 120.9, 124.1 (d), 127.9 (d), 129.4, 130.2, 136.3, 138.2, 146.0, and 150.8 (Ar-$C$).

$^{31}P\{^1H\}$ **NMR** ($D_8$-THF, 162 MHz, 298 K): $\delta = 62.5$ (s, $^iPr_2P$-CO).

$^{29}Si\{^1H\}$ **NMR** ($D_8$-THF, 99 MHz, 298 K): $\delta = -28.0$.

$^9$**Be NMR** ($D_8$-THF, 58 MHz, 298 K), $\delta = 8.2$.

**MS/LIFDI-HRMS** found (calcd.) $m/z$: 1052.6117 (1052.6103) for $[M]^+$.

## Data availability

Crystallographic Information Files have been deposited in the Cambridge Crystallographic Data Centre (deposition numbers: 2101682−2101690) and can be accessed free of charge from The Cambridge Crystallographic Data Centre via www.ccdc.cam.ac.uk/data_request/cif. All information supporting the findings in this study are available either within the article, its Supplementary Information, or are available from the corresponding author upon reasonable request.

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

## Acknowledgements
T.J.H. is grateful for the receipt of a Liebig Stipendium of the Fonds der Chemischen Industrie, and for the receipt of funding from the Technical University Munich as a TUM Junior Fellow. He would also like to thank Prof. F. Krauss and Dr. M. Buchner for the generous gift of elemental beryllium, P. Keil for aiding in the measurement of LIFDI-MS and IR spectra, and C. Fajman and T. Restle for aiding in running and analysing Power X-ray diffraction experiments. T.S. thanks the University of Alabama and the Office of Information Technology for providing high performance computing resources and support. This computational work was made possible in part by a grant of high-performance computing resources and technical support from the Alabama Super-computer Authority.

## Author contributions
T.J.H. designed and planned the project, and carried out all experimental work. T.S. performed computational investigations.

## Funding

## Competing interests
The authors declare no competing interests.
