## [Peer Review File · Nature Communications]

Accessing the Main-Group Metal Formyl Scaffold through CO-Activation in Beryllium Hydride ComplexesREVIEWER COMMENTS

Reviewer #1 (Remarks to the Author):

The manuscript by Hadlington and Szilvasi reports the activation of Be-H bonds by CO, which is observed for the first time. The observation of such reactivity is a significant step in the advancement of beryllium chemistry, which has potential for hydrogen storage.

The experimental work appear to be well described, with structures well characterised with X-ray studies, NMR and IR spectra, and high-res MS. The authors also carried out a theoretical analysis with DFT, which I will discuss in more detail below.

In the introduction, the last part on p3 describes results and conclusions. Maybe that is reasonable for a short communication, however I would suggest re-working this part of the manuscript to leave results and conclusions for the results and conclusion sections.

The DFT computational study employs reasonable methods for the mechanistic study. It is perhaps surprising that different functionals are employed for the geometry optimization and single-point energies, however the impact would be small. The mechanistic study is well described, with IRC used to confirm the mechanism. It is pleasing to see all Cartesian coordinates included in the ESI.

The authors have referenced previous Be-H insertion reactivity of Hill and co-workers (Ref 22-23), however it would be relevant to cite a related computational study of the mechanism (Organometallics 2013, 32, 6209) given the current manuscript describes a combined synthesis and computational study.

The principal issue with the computational studies is the heavy reliance on NBO. A number of authors have highlighted the failure of the NBO method to provide a chemically intuitive description for Be as it only considers the 2s AO and omits the important 2p AO (Organometallics 2020, 39, 3224; Angew. Chem. Int. Ed. 2019, 58, 1732; Angew. Chem. Int. Ed. 2021, 60, 9407). A relevant example of this is a computational study of BeO rings (Dalton Trans., 2018, 47, 12633), where similar results are obtained as in the present study. The failure to account for the Be 2p AO leads to (unreasonable) purely ionic structures. That leads to very large NPA charges - here Hirschfeld CM5 charges are a better alternative. In place of NBO, it is likely that QTAIM would yield a more reliable analysis.

The bonding in the planar core [Be₂C₂O₂] in compound 11 is fascinating and unexpected. The authors are encouraged to explore this in more detail as described above, with analysis beyond NBO and MO calculations.

Reviewer #2 (Remarks to the Author):

The manuscript by Hadlington and coworkers reports an amido(w/pendant phosphine)-beryllium hydride monomer and dimer that react with both CO and CO₂. These reactions led to the formation of a formate-bridged beryllium dimer and a dimeric beryllium formyl complex. Our knowledge of the synthetic and reaction chemistry of molecular beryllium hydrides is extremely narrow with only a handful of structurally characterized compounds in the literature. Such compounds, as observed here via bubbling in perfluorinated oil, are very difficult to synthesize and successfully characterize. Certainly, the CO and CO₂ chemistry presented is novel, affording mono-insertion into Be-H. The experimentally challenging work is both thorough and novel. The manuscript is written at a high scholarly level, including the supporting information, and I strongly support publication in Nature Communications with minor revisions. For me, this is an easy case and an excellent contribution to the field of organoberyllium chemistry! This work also crosses boundaries, being important not only

for s-block chemistry, but also general organometallic chemistry.

Comments and recommended improvements:

- 1) In line 118, a reference is missing for the NHC-Be complex bond distance.
- 2) Regarding lines 124-126, the broadening is quite standard for ^9Be NMR: see comments and related material *Inorg. Chem.* 1997, 36, 21, 4688–4696; *J. Am. Chem. Soc.* 2004, 126, 44, 14651–14658; *Zeitschrift für Naturforschung B* 2020, 75, 459.
- 3) In the supporting information the Be-H dimer (Figure S28) appears to have two peaks, or it has a sharp shoulder on the peak at 6.90 ppm. What is this? Also, there appears to be a smaller peak growing in around 11 ppm.

Reviewer #3 (Remarks to the Author):

I've been asked to review the X-ray crystallography aspects of the manuscript titled "Accessing the Main-Group Metal Formyl Scaffold through CO-Activation in Beryllium Hydride Complexes" so my comments are focussed on this area. Overall I find this work fascinating as not many chemists are brave enough to work with Beryllium at all, and the authors have made some lovely compounds showing some unusual reactivity towards CO and CO₂. The authors report 9 new crystal structures (compounds 4-12) and have provided complete cifs and checkcifs for these compounds. Overall the data is of good quality and the data-sets have been refined to a good standard and in each case the data supports the authors claims as to chemical structure and geometry. Typically a problem with X-ray diffraction is reliably locating metal-hydride ligands but as the metal of interest here is electron poor beryllium, there are no issues locating the hydride ligands, with the hydrides found in the electron density map in each case with no constraints or restraints required in any of the refinements. Although the overall quality is good there are a few technical details in the cifs that should be addressed before publication.

- 1) All the supplied cifs are missing the following fields for software used: `_computing_cell_refinement`,

`_computing_data_collection`
`_computing_data_reduction`
`_computing_structure_solution` "

Please add this information along with the following :

`_atom_sites_solution_primary` ? (Usually direct when using ShelXS)

`_atom_sites_solution_secondary` ? (Usually difmap when using ShelXS)

- 2) Compounds 4, 6 and 7 each have disagreeable reflections that should be omitted from the final refinement cycles (See various checkcif alert numbers PLAT918 and PLAT934). For 4 it is hkl 6 7 9, for 6: 2 -9 4; for 7: -10 -4 6. These are likely biasing your weighting schemes and the refinement will benefit from omitting them.

- 3) The formula for 5 in the cif is missing two hydrogens. Please update formula and refine to give correct formula/density/F(000) values etc.

- 4) The moiety formula for 7 and 9 match the sum formula and don't take into account the lattice solvent. E.g the moiety formula for 7 should be C₇₄ H₈₀ Be₂ N₂ P₂ Si₂, 2(C₇ H₈)

- 5) The unit cell for 10 is monoclinic but the beta angle is <90° (71.00°). The standard setting is to have the beta angle >90, is there a reason for this?

- 6) Cifs do not contain X-H bond distances and uncertainties. See point 8 for more information below. Some other comments on the manuscript and supporting information:

- 7) Figures 2-4 and S74 : Many stated bond distances do not match those from the cifs. e.g. for 6, the Be1-Be1' distance is given as 2.123(6) but the cif gives it as 2.124(8). For 5, the Be1-O1 distance is given as 1.633(4) but is 1.633(3) in the cif. For 9 the Be1-C32 distance is given as 1.837(5) but in the cif it is 1.800(5). There are too many to list out completely, can the authors correct and fix these.

- 8) In the figure 2 and 3 captions Be-H distances are given, some with uncertainties and some without. How were these distances and uncertainty determined? In the cifs no X-H bond distances are given. These can be accurately calculated using ShelXl by adding a command to the .ins file ("Bond \$H"). This will include X-H bond parameters in the output cif file that will give you accurate bond distance and uncertainties. Other ways of measuring the distance and uncertainty likely don't do a proper variance-co-variance matrix calculation so will give inaccurate results.
- 9) Line 197-198: "and the formyl C-O bonds long ($d_{CO} = 1.24(3) \text{ \AA}$)," I think this is a typo as the distance is $1.424(3) \text{ \AA}$ in the cif. (The distance in the figure 4 caption is also incorrect but see comment 7)
- 10) The spelling of fluxional (vs fluctional) varies through the manuscript.
- 11) SI: line 208, Schlenk should be Schlenk flask or Schlenk tube.
- 12) SI: Table S1 the space group for 4 should be P-1 and not triclinic.

Reviewer #1 (Remarks to the Author):

The manuscript by Hadlington and Szilvasi reports the activation of Be-H bonds by CO, which is observed for the first time. The observation of such reactivity is a significant step in the advancement of beryllium chemistry, which has potential for hydrogen storage.

The experimental work appear to be well described, with structures well characterised with X-ray studies, NMR and IR spectra, and high-res MS. The authors also carried out a theoretical analysis with DFT, which I will discuss in more detail below.

In the introduction, the last part on p3 describes results and conclusions. Maybe that is reasonable for a short communication, however I would suggest re-working this part of the manuscript to leave results and conclusions for the results and conclusion sections.

We have considerably shortened the closing statement of the Introduction, in line with these comments. We hope this makes the transition to the 'Results and Discussion' part smoother.

The DFT computational study employs reasonable methods for the mechanistic study. It is perhaps surprising that different functionals are employed for the geometry optimization and single-point energies, however the impact would be small. The mechanistic study is well described, with IRC used to confirm the mechanism. It is pleasing to see all Cartesian coordinates included in the ESI.

The authors have referenced previous Be-H insertion reactivity of Hill and co-workers (Ref 22-23), however it would relevant to cite a related computational study of the mechanism (Organometallics 2013, 32, 6209) given the current manuscript describes a combined synthesis and computational study.

This computational study has been briefly discussed in the introduction to Be-H chemistry.

The principal issue with the computational studies is the heavy reliance on NBO. A number of authors have highlighted the failure of the NBO method to provide a chemically intuitive description for Be as it only considers the 2s AO and omits the important 2p AO (Organometallics 2020, 39, 3224; Angew. Chem. Int. Ed. 2019, 58, 1732; Angew. Chem. Int. Ed. 2021, 60, 9407). A relevant example of this is a computational study of BeO rings (Dalton Trans., 2018, 47, 12633), where similar results are obtained as in the present study. The failure to account for the Be 2p AO leads to (unreasonable) purely ionic structures. That leads to very large NPA charges - here Hirschfeld CM5 charges are a better alternative. In place of NBO, it is likely that QTAIM would yield a more reliable analysis.

We thank the reviewer for pointing out this issue of NBO analysis. In the revised version, we also analyze QTAIM and Hirschfeld CM5 charges. Values for these analyses are added in Supporting Information (Supporting Table 4), and the following paragraph has been added to the main text (with relevant references, namely 39-41):

"However, the weakness of NBO analysis for Be compounds has been noted in previous computational works because it does not consider important 2p atomic orbitals, and only 2s atomic orbitals, thus overpredicting ionic character of Be bonds. We have therefore also calculated the extended Hirshfeld method CM5 charges, Electron Density, Laplacian of Electron Density, Kinetic Energy Density, Potential Energy Density, Total Electronic Energy Density, and Ellipticity at the Be related Bond Critical Points of

*the central Be₂C₂O₂ moiety in **11'** (see Supplementary Table 4). We found that CM5 charge shows a considerably more charged balanced, less ionic picture compared to NPA, having Be and O charges of +0.63 and -0.41, respectively. Topological analysis of the electron density shows positive values for the Laplacian of electron density at the Bond Critical Points (BCP), which is a common error, therefore we analysed the more general total electron density descriptor. The total electron density is negative at both the Be-C (-0.016) and Be-O (-0.017) BCPs, suggesting covalent character for these interactions. Intriguingly, the ellipticity of the Be-C bond at the BCP is 0.12, which even suggests slight double bond character. Overall, this gives an overall more covalent picture for the Be-O bonding in **11'**, and maintains a dative picture for the Be-C bonding interactions."*

The bonding in the planar core [Be₂C₂O₂] in compound **11** is fascinating and unexpected. The authors are encouraged to explore this in more detail as described above, with analysis beyond NBO and MO calculations.

Reviewer #2 (Remarks to the Author):

The manuscript by Hadlington and coworkers reports an amido(w/pendant phosphine)-beryllium hydride monomer and dimer that react with both CO and CO₂. These reactions led to the formation of a formate-bridged beryllium dimer and a dimeric beryllium formyl complex. Our knowledge of the synthetic and reaction chemistry of molecular beryllium hydrides is extremely narrow with only a handful of structurally characterized compounds in the literature. Such compounds, as observed here via bubbling in perfluorinated oil, are very difficult to synthesize and successfully characterize. Certainly, the CO and CO₂ chemistry presented is novel, affording mono-insertion into Be-H. The experimentally challenging work is both thorough and novel. The manuscript is written at a high scholarly level, including the supporting information, and I strongly support publication in Nature Communications with minor revisions. For me, this is an easy case and an excellent contribution to the field of organoberyllium chemistry! This work also crosses boundaries, being important not only for s-block chemistry, but also general organometallic chemistry.

We'd like to thank this reviewer for their kind comments regarding this work. It was indeed quite challenging, but rewarding to observe this broadly un-investigated chemistry.

Comments and recommended improvements:

1) In line 118, a reference is missing for the NHC-Be complex bond distance.

References have been added here.

2) Regarding lines 124-126, the broadening is quite standard for ⁹Be NMR: see comments and related material Inorg. Chem. 1997, 36, 21, 4688–4696; J. Am. Chem. Soc. 2004, 126, 44, 14651–14658; Zeitschrift für Naturforschung B 2020, 75, 459.

Indeed, we're aware of the typically broad signals for ⁹Be NMR given the quadrupolar nature of the element. We've added a note in the references at that point in the text.

3) In the supporting information the Be-H dimer (Figure S28) appears to have two peaks, or it has a sharp

shoulder on the peak at 6.90 ppm. What is this? Also, there appears to be a smaller peaks growing in around 11 ppm.

We believe this is due to a small amount of free ligand. In the stack plot shown we have compared the ^{31}P NMR of the complex $[\text{i}^{\text{PrPh}}\text{DippNBeH}]_2$ with protonate ligand, $\text{i}^{\text{PrPh}}\text{DippNH}$, which was generated in-situ through quenching the potassium salt with $[\text{Et}_3\text{NHCl}]$. Of course, signal broadening for $[\text{i}^{\text{PrPh}}\text{DippNBeH}]_2$ 'accentuates' the small but sharp free ligand peak. We are presently unsure as to the source of the small peak at 11 ppm, but believe this could be due to the some decomposition to the hydroxide compound, $\text{i}^{\text{PrPh}}\text{DippNBeOH}$, due to reaction with moisture adsorbed to the NMR tube which is known to become more 'available' upon extended heating of a sample (note the ^{31}P NMR spectrum for $[\text{i}^{\text{PrPh}}\text{DippNBeH}]_2$ had to be collected at high temperature due to very poor solubility). We're presently attempting to directly access the beryllium hydroxide complex, to see if our superstitions are accurate!

Reviewer #3 (Remarks to the Author):

I've been asked to review the X-ray crystallography aspects of the manuscript titled "Accessing the Main-Group Metal Formyl Scaffold through CO-Activation in Beryllium Hydride Complexes" so my comments are focussed on this area. Overall I find this work fascinating as not many chemists are brave enough to work with Beryllium at all, and the authors have made some lovely compounds showing some unusual reactivity towards CO and CO₂. The authors report 9 new crystal structures (compounds 4-12) and have provided complete cifs and checkcifs for these compounds. Overall the data is of good quality and the data-sets have been refined to a good standard and in each case the data supports the authors claims as to chemical structure and geometry. Typically a problem with X-ray diffraction is reliably locating metal-hydride ligands but as the metal of interest here is electron poor beryllium, there are no issues locating the hydride ligands, with the hydrides found in the electron density map in each case with no constraints or restraints required in any of the refinements. Although the overall quality is good there are a few technical details in the cifs that should be addressed before publication.

We very much appreciate the kind comments regarding our 'brave' work. We hope the changes and additions we've made to the crystal data bring them up to the required standard!

1) All the supplied cifs are missing the following fields for software used:

_computing_cell_refinement, _
_computing_data_collection
_computing_data_reduction
_computing_structure_solution "

Please add this information along with the following :

_atom_sites_solution_primary ? (Usually direct when using ShelXS)
_atom_sites_solution_secondary ? (Usually difmap when using ShelXS)

These details have been added for all compounds.

2) Compounds **4**, **6** and **7** each have disagreeable reflections that should be omitted from the final refinement cycles (See various checkcif alert numbers PLAT918 and PLAT934). For **4** it is hkl 6 7 9, for **6**: 2 -9 4; for **7**: -10 -4 6. These are likely biasing your weighting schemes and the refinement will benefit from omitting them.

Not quite sure how we overlooked these before, but they've now all been excluded from the final refinements.

3) The formula for **5** in the cif is missing two hydrogens. Please update formula and refine to give correct formula/density/F(000) values etc.

This has been corrected, and the data updated.

4) The moiety formula for **7** and **9** match the sum formula and don't take into account the lattice solvent. E.g the moiety formula for **7** should be C₇₄ H₈₀ Be₂ N₂ P₂ Si₂, 2(C₇ H₈)

*This has been corrected, although the non-integer value of 0.33(C₅H₁₂) in the moiety formula of **9** flags a G level error for non-matching moiety and sum formulas (although this in our opinion is not a real problem).*

5) The unit cell for **10** is monoclinic but the beta angle is <90° (71.00°). The standard setting is to have the beta angle >90, is there a reason for this?

*Indeed, we thought the same, but this is the value given during data reduction and scaling. For the powder sample, using the inverse value (109°) gives largely the same predicted diffractogram, but with 'flipped' intensities for two or three major reflections. Correcting this to the 71° angle gave a predicted diffractogram which matched that for the diffractogram of powdered crystals of **10** (Fig. S76). As such, the angle of 71° was not corrected.*

6) Cifs do not contain X-H bond distances and uncertainties. See point 8 for more information below.

This has been corrected with the BOND \$H command.

Some other comments on the manuscript and supporting information:

7) Figures 2-4 and S74 : Many stated bond distances do not match those from the cifs. e.g. for **6**, the Be1-Be1' distance is given as 2.123(6) but the cif gives it as 2.124(8). For **5**, the Be1-O1 distance is given as 1.633(4) but is 1.633(3) in the cif. For **9** the Be1-C32 distance is given as 1.837(5) but in the cif it is 1.800(5). There are too many to list out completely, can the authors correct and fix these.

All bond lengths and angles have been double checked against their respective (new) cifs. They should all now be in keeping.

8) In the figure 2 and 3 captions Be-H distances are given, some with uncertainties and some without. How were these distances and uncertainty determined? In the cifs no X-H bond

distances are given. These can be accurately calculated using ShelXL by adding a command to the .ins file ("Bond \$H"). This will include X-H bond parameters in the output cif file that will give you accurate bond distance and uncertainties. Other ways of measuring the distance and uncertainty likely don't do a proper variance-co-variance matrix calculation so will give inaccurate results.

As mentioned above, the BOND \$H command has been implanted, reading X-H distances to the cif. For Be-H moieties the H atoms were freely refined, and presumably due to data quality differences the accuracy of these values differ considerably, in some cases no uncertainty been given (I've experienced this a number of times for main-group hydride systems). As such, uncertainties for these bonds have been removed from the text.

9) Line 197-198: "and the formyl C-O bonds long ($d_{CO} = 1.24(3) \text{ \AA}$)," I think this is a typo as the distance is $1.424(3) \text{ \AA}$ in the cif. (The distance in the figure 4 caption is also incorrect but see comment 7)

This has been corrected.

10) The spelling of fluxional (vs fluctional) varies through the manuscript.

This has been changed to 'fluctional' throughout

11) SI: line 208, Schlenk should be Schlenk flask or Schlenk tube.

This has been changed.

12) SI: Table **S1** the space group for **4** should be P-1 and not triclinic.

This has been changed.

REVIEWER COMMENTS

Reviewer #1 (Remarks to the Author):

I was reviewer #1 for the original submission. The authors have satisfactorily addressed my comments and questions, and in my opinion the paper is ready for publication in Nat. Comms. This manuscript is well-written, the work is thorough and novel, and altogether this makes a very strong contribution to the development of organoberyllium chemistry.

Reviewer #2 (Remarks to the Author):

I read the authors comment regarding what I originally thought was the ^9Be NMR spectrum of the Be-H dimer, which is why I was confused seeing the extra peaks. The response clarifies that that is instead the ^{31}P NMR spectrum, which makes more sense, but that means there is a mistake in the labelling. The authors need to check the figure caption for "Supplementary Figure 28." which says that the above figure is the ^9Be NMR spectrum. However, it does not match with the label that is on the actual NMR spectrum. It appears that the same mistake in the labelling exist for "Supplementary Figure 29", which is the ^{13}C NMR spectrum but is labelled ^9Be NMR spectrum. Regarding the impurities I will leave it up to the authors whether they want to denote these in the supporting information. I always think it is useful for the field and general reader to make a note in the caption or on the spectrum when these experimental challenges arise. This helps others who may need to refer to this work so they are not confused by the additional peaks. It is certainly ok to state "unidentified impurity" or include a note with your speculation based on the experiments. After the minor corrections in labelling are made this work is suitable for publication and I look forward to reading this manuscript in its final form.

Reviewer #3 (Remarks to the Author):

Thank you to the authors for responded to my comments satisfactorily, I have one comment based on their statement:

"As mentioned above, the BOND \$H command has been implanted, reading X-H distances to the cif. For Be-H moieties the H atoms were freely refined, and presumably due to data quality differences the accuracy of these values differ considerably, in some cases no uncertainty been given (I've experienced this a number of times for main-group hydride systems). As such, uncertainties for these bonds have been removed from the text."

I'm not sure if they were there before and I didn't notice or they have crept in but the Hydrogen atoms in 6, 7, 8, 9 each have an AFIX command associated with them (either AFIX 2 or AFIX 3), which is constraining one or more parameters.

When AFIX 2 is used (e.g. compound 9), this allows the coordinates of the H to refine, but fixes the U to ride the parent atom; this gives an uncertainty to the distance.

Whereas when AFIX 3 is used (e.g. compound 6) the geometry rides the parent atom and freely refines the U parameter, so no uncertainty is generated for the Be-H distance.

I would refine with no AFIX at all and see if you get a stable refinement, this would give you the uncertainties. If its not stable, use AFIX 2 rather than AFIX 3 to aid refinement, as this will give you uncertainties on your Be-H distance.

Response to referee comments:

Reviewer #1 (Remarks to the Author):

I was reviewer #1 for the original submission. The authors have satisfactorily addressed my comments and questions, and in my opinion the paper is ready for publication in Nat. Comms. This manuscript is well-written, the work is thorough and novel, and altogether this makes a very strong contribution to the development of organoberyllium chemistry.

Reviewer #2 (Remarks to the Author):

I read the authors comment regarding what I originally thought was the ^9Be NMR spectrum of the Be-H dimer, which is why I was confused seeing the extra peaks. The response clarifies that that is instead the ^{31}P NMR spectrum, which makes more sense, but that means there is a mistake in the labelling. The authors need to check the figure caption for "Supplementary Figure 28." which says that the above figure is the ^9Be NMR spectrum. However, it does not match with the label that is on the actual NMR spectrum. It appears that the same mistake in the labelling exist for "Supplementary Figure 29", which is the ^{13}C NMR spectrum but is labelled ^9Be NMR spectrum. Regarding the impurities I will leave it up to the authors whether they want to denote these in the supporting information. I always think it is useful for the field and general reader to make a note in the caption or on the spectrum when these experimental challenges arise. This helps others who may need to refer to this work so they are not confused by the additional peaks. It is certainly ok to state "unidentified impurity" or include a note with your speculation based on the experiments. After the minor corrections in labelling are made this work is suitable for publication and I look forward to reading this manuscript in its final form.

Miscellaneous peaks in five spectra (Supplementary Figures 9, 10, 17, 25, 28) have been highlighted with asterisk, and their origin noted (if known). Further, the figure captions have been corrected for Supplementary Figures 20, 28, and 29.

Reviewer #3 (Remarks to the Author):

Thank you to the authors for responded to my comments satisfactorily, I have one comment based on their statement:

"As mentioned above, the BOND \$H command has been implanted, reading X-H distances to the cif. For Be-H moieties the H atoms were freely refined, and presumably due to data quality differences the accuracy of these values differ considerably, in some cases no uncertainty been given (I've experienced this a number of times for main-group

hydride systems). As such, uncertainties for these bonds have been removed from the text."

I'm not sure if they were there before and I didn't notice or they have creeped in but the Hydrogen atoms in 6, 7, 8, 9 each have an AFIX command associated with them (either AFIX 2 or AFIX 3), which is constraining one or more parameters.

When AFIX 2 is used (e.g. compound 9), this allows the coordinates of the H to refine, but fixes the U to ride the parent atom; this gives an uncertainty to the distance.

Whereas when AFIX 3 is used (e.g. compound 6) the geometry rides the parent atom and freely refines the U parameter, so no uncertainty is generated for the Be-H distance.

I would refine with no AFIX at all and see if you get a stable refinement, this would give you the uncertainties. If its not stable, use AFIX 2 rather than AFIX 3 to aid refinement, as this will give you uncertainties on your Be-H distance.

Indeed, the AFIX 2/3 values were implemented in the original submission, which we hadn't linked to the inaccuracy in bond Be-H bond lengths, but this appears to have solved the issue. All Be-H moieties (and B-H moieties in 5) have been refined with AFIX 0, improving the accuracy of these bond distances, and giving error values in all cases. These errors are now included in the main text and supporting information. The new CIFs are included in this submission, and have been deposited in the CCDC under the same deposition numbers given previously for those compounds. R-values have also been cross checked and any minor changes due to the new refinements are updated in the Supporting Information.

REVIEWER COMMENTS

Reviewer #3 (Remarks to the Author):

Thank you to the authors for fixing the problems addressed in my last comments. I think the cifs are now complete and ready for publication. I have scanned the Supplementary tables and there are still a couple of very minor discrepancies as a result of the re-refined cifs. e.g. compound 6, the listed value for R1 is 0.0693 but in the cif it is 0.0694. Could the authors and/or editors have a final check that these all match before publication?

Response to Reviewers

Thank you to the authors for fixing the problems addressed in my last comments. I think the cifs are now complete and ready for publication. I have scanned the Supplementary tables and there are still a couple of very minor discrepancies as a result of the re-refined cifs. e.g. compound 6, the listed value for R1 is 0.0693 but in the cif it is 0.0694. Could the authors and/or editors have a final check that these all match before publication?

The crystallography tables have been check against all CIFs, and no discrepancies could be found.